# A Gata3–Mafb transcriptional network directs post-synaptic differentiation in synapses specialized for hearing

Wei-Ming Yu[1], Jessica M Appler[1†], Ye-Hyun Kim[2†], Allison M Nishitani[1], Jeffrey R Holt[2], Lisa V Goodrich[1]*

[1]Department of Neurobiology, Harvard Medical School, Boston, United States; [2]Department of Otolaryngology and FM Kirby Neurobiology Center, Children's Hospital Boston, Harvard Medical School, Boston, United States

**Abstract** Information flow through neural circuits is determined by the nature of the synapses linking the subtypes of neurons. How neurons acquire features distinct to each synapse remains unknown. We show that the transcription factor Mafb drives the formation of auditory ribbon synapses, which are specialized for rapid transmission from hair cells to spiral ganglion neurons (SGNs). Mafb acts in SGNs to drive differentiation of the large postsynaptic density (PSD) characteristic of the ribbon synapse. In *Mafb* mutant mice, SGNs fail to develop normal PSDs, leading to reduced synapse number and impaired auditory responses. Conversely, increased Mafb accelerates synaptogenesis. Moreover, Mafb is responsible for executing one branch of the SGN differentiation program orchestrated by the Gata3 transcriptional network. Remarkably, restoration of Mafb rescues the synapse defect in *Gata3* mutants. Hence, Mafb is a powerful regulator of cell-type specific features of auditory synaptogenesis that offers a new entry point for treating hearing loss.

*For correspondence: Lisa_Goodrich@hms.harvard.edu

†These authors contributed equally to this work

Competing interests: The authors declare that no competing interests exist.

## Introduction

Synapses are the basic building blocks for the diverse types of neural circuits that mediate perception and behavior. How information is processed and transmitted is determined both by the identity of the neurons in the circuit and the nature of their synaptic connections. Indeed, synapses exhibit a range of morphologies and functions, varying both the nature of the signal and its strength. Recent efforts to unravel the molecular basis of synaptic specificity have begun to define the mechanisms that match pre- and post-synaptic partners and thereby establish the overall wiring pattern. These studies suggest that the fate of each partner is determined by cell-type specific transcription factors, which activate expression of a unique combination of cell-surface receptors to permit adhesion only between appropriate partners (*Polleux et al., 2007*; *Sanes and Yamagata, 2009*). In contrast, we still know very little about how each partner is subsequently instructed to develop the appropriate type of pre- or post-synaptic specialization required at that synapse. Although several broadly acting transcription factors have been shown to control synapse number, the identity of the transcription factors that might direct cell-type specific features of synaptic development remains elusive.

Among the most specialized synapses in the nervous system are those of the auditory system that transmit sound information from inner hair cells (IHCs) to spiral ganglion neurons (SGNs). As the sole link between IHCs and the brain, SGNs encode the frequency, intensity, and timing of all acoustic stimuli and communicate this information rapidly and faithfully to the brain (*Appler and Goodrich, 2011*). To achieve the necessary speed, SGNs receive information from IHCs via ribbon synapses that are specialized for signaling with high temporal precision (*Khimich et al., 2005*; *Glowatzki et al., 2008*;

**eLife digest** Different types of neurons in the nervous system communicate with each other through different types of synapses. In the auditory system, for example, it is essential for the timing of signals to be preserved as they are sent from the ear to the brain, and this places special demands on the synapses in this part of the nervous system. In particular, the ribbon synapses that are found between the inner hair cells of the ear, which convert sound waves into neural signals, and the neurons of the spiral ganglion in the cochlea, which carry information about the frequency, intensity and timing of sounds to the brain, can transmit signals with remarkable fidelity.

Little is known about the mechanisms by which synapses become specialized for particular functions. Previous work has suggested that a protein called Gata3 is important for the development of the neurons and synapses in the spiral ganglion, including ribbon synapses. Gata3 is a transcription factor that controls the expression of a wide range of genes that are involved in the auditory systems, including genes that are expressed as other transcription factors.

Yu et al. used transgenic mice to explore what happened when one of these transcription factors, Mafb, was missing from neurons in the spiral ganglion. The results showed that ribbon synapses did not form when Mafb was absent, which meant that they were unable to respond normally to sounds. Yu et al. also studied mice in which Gata3 was absent: normally Mafb would not be present in these mice, but when genetic techniques were used to force the expression of the gene for Mafb, ribbon synapses were formed. As well as revealing a molecular pathway by which synapses become specialized for rapid and accurate transmission of auditory information, these findings might lead to new approaches to treating hearing loss in humans.

*Buran et al., 2010*; *Safieddine et al., 2012*). Unlike conventional synapses, ribbon synapses contain an electron-dense multi-protein ribbon structure, which tethers a large pool of readily releasable vesicles (*Sterling and Matthews, 2005*; *Goutman and Glowatzki, 2007*; *Frank et al., 2010*), thereby enabling fast and sustained release of glutamate from the IHCs (*Glowatzki and Fuchs, 2002*). Likewise, SGNs respond with remarkable speed due to a large postsynaptic density (PSD) containing abundant clusters of AMPA-type receptors (*Matsubara et al., 1996*). Since the features of all sound stimuli are captured by the pattern of synaptic transmission between IHCs and SGNs, developing the correct number and types of synapses is critical for hearing. Indeed, auditory synaptopathies underlie several forms of hearing loss (*Moser et al., 2013*), and ribbon synapses are particularly vulnerable to acoustic overexposure (*Kujawa and Liberman, 2009*; *Lin et al., 2011*).

Despite the importance of ribbon synapses to the sense of hearing, we have only a rudimentary understanding of how this synapse acquires its specialized structures and properties during development. In mice, ribbon synapses begin to develop perinatally (*Sobkowicz et al., 1982*; *Nemzou et al., 2006*; *Sendin et al., 2007*; *Marrs and Spirou, 2012*). Presynaptic ribbons are initially distributed throughout the IHC cytoplasm and then gradually localize to the basolateral surface to form an immature synapse, with clustered ribbons in large active zones (*Sobkowicz et al., 1982*; *Sendin et al., 2007*). In parallel, SGN neurites reach IHCs and begin to elaborate post-synaptic terminals, characterized by increased clustering of GluR2/R3 AMPA receptors in the nascent PSD (*Huang et al., 2012*). Ribbon number peaks at the end of the first postnatal week, followed by a period of pruning and refinement. By the onset of hearing (postnatal day 12 [P12] in mice), ribbon number has declined to adult levels and each ribbon is anchored opposite a specialized large, well-defined PSD in the SGN terminal (*Sobkowicz et al., 1982*; *Nemzou et al., 2006*; *Sendin et al., 2007*). In mature animals, the size of the PSD correlates with the size and position of the ribbon (*Liberman et al., 2011*), indicating that pre- and post-synaptic development must be tightly coordinated. Indeed, without SGN afferent terminals, ribbon development in IHCs is impaired (*Sobkowicz et al., 1986*), and conversely, afferent terminals are not stabilized in mutants lacking normal ribbons (*Sheets et al., 2011*). However, the molecular mechanisms that allow SGNs to form these unusual connections with IHCs remain a mystery.

Efforts to understand how SGNs acquire their auditory-specific features have established an important role for the Gata3 transcription factor, which is expressed in SGNs as soon as they can be identified (*Lawoko-Kerali et al., 2004*) and then maintained throughout differentiation (*Appler et al., 2013*;

*Duncan and Fritzsch, 2013*; *Luo et al., 2013*). Upon deletion of *Gata3*, cochlear wiring is severely disrupted and SGNs fail to express a cohort of auditory-specific genes (*Appler et al., 2013*; *Duncan and Fritzsch, 2013*), including additional transcription factors that may work downstream of Gata3 to direct specific features of SGN differentiation. One particularly intriguing putative downstream effector is the basic leucine-zipper transcription factor Mafb. Mafb belongs to the family of large Maf factors, which activate cell-type specific programs of gene expression essential for terminal differentiation of a wide variety of cell types, including pancreatic β cells, kidney podocytes, and macrophages (*Moriguchi et al., 2006*; *Kataoka, 2007*; *Hang and Stein, 2011*). *Mafb* was first identified as the genetic cause of the *kreisler* mutation (*kr*) in mice (*Cordes and Barsh, 1994*). In *kr* mutants, rhombomeres 5 and 6 are not properly specified, and the otic vesicle, which develops adjacent to these structures, fails to acquire its normal morphology. Although its function in the nervous system remains unclear, *Mafb* is also required in the pre-Bötzinger complex in the hindbrain (*Blanchi et al., 2003*): respiratory circuits are unable to mediate normal breathing in *Mafb* mutants, which die from central apnea at birth. Unfortunately, the early lethality and inner ear defects in null mutants have prevented analysis of Mafb's role in the assembly and function of auditory circuits.

In this study, through genetic analysis in mice, we demonstrate that Mafb is critical for the ability of SGNs to form the specialized contacts necessary for the sense of hearing. Mafb is enriched in SGNs during synaptogenesis. Further, deletion of Mafb from SGNs disrupts post-synaptic differentiation, leading to an overall reduction in ribbon synapse number and hence impaired auditory responses. Conversely, restoration of Mafb is sufficient to rescue the synaptic defect seen in *Gata3* mutant mice. These studies establish Mafb as a lineage-specific intrinsic regulator in SGNs for post-synaptic specification of ribbon synapses and a powerful effector within a broader Gata3 network during auditory circuit assembly.

## Results

### Mafb expression peaks during synaptogenesis in the cochlea

The first clue that Mafb might serve as a lineage-specific regulator of SGN terminal differentiation came from microarray comparisons of SGNs and the closely related vestibular ganglion neurons (VGNs), which revealed a dramatic SGN-specific increase in *Mafb* expression during late embryogenesis (*Lu et al., 2011*). SGN neurogenesis occurs along a base to apex gradient, with most SGNs exiting the cell cycle by E12.5 and extending projections to reach the organ of Corti by E15.5 (*Matei et al., 2005*; *Koundakjian et al., 2007*). In situ hybridization confirmed that *Mafb* transcription initiates in parallel with the end of neurogenesis, with expression evident in the basal to middle regions of the cochlea by E13.5 (*Figure 1B*). Protein was not yet present at this stage (*Figure 1C,C'*). Two days later, Mafb protein could be detected in SGNs in the basal and middle turns (*Figure 1D*) and by E16.5, Mafb protein was localized to SGN nuclei along the entire length of the cochlea (*Figure 1E*). High levels of Mafb persisted during early postnatal stages (*Figure 1F,J*) and peaked at P6 (*Figure 1G,J*). Afterward, Mafb was gradually translocated from the nucleus to the cytoplasm, and only a few SGNs maintained strong nuclear expression by P10 (*Figure 1H*, arrows). After the onset of hearing, Mafb was still present at low levels in the SGN cytoplasm (*Figure 1I,J*). At all stages, Mafb expression was limited to SGNs, with no protein detected in VGNs (data not shown) or other cell types of the inner ear. Hence, Mafb defines the SGN subset of inner ear neurons and is expressed throughout auditory synaptogenesis, which begins around perinatal stages and lasts for the first two postnatal weeks, peaking around P6 (*Sobkowicz et al., 1982*; *Huang et al., 2012*; *Safieddine et al., 2012*).

### Mafb is not required for early SGN development and differentiation

A role for Mafb in differentiating SGNs has not been investigated due to the early lethality and severe inner ear malformations of null mutants (*Choo et al., 2006*). To bypass these constraints, we generated a conditional *Mafb* allele (*Mafbflox*) (*Figure 2A,B*). *Mafbflox/flox* homozygotes are fertile and viable, with normal hearing and no gross anatomical defects. However, after germline Cre-mediated recombination, homozygotes produced no Mafb protein (*Figure 2C*) and exhibited the same phenotypes as previously published null mutants (*Figure 2D–F*), which die perinatally with defective respiratory rhythmogenesis, renal dysgenesis, and cystic inner ears (*Sadl et al., 2002*; *Blanchi et al., 2003*; *Choo et al., 2006*; *Moriguchi et al., 2006*). Hence, the *Mafbflox* allele provides an effective tool for conditional control of Mafb function.

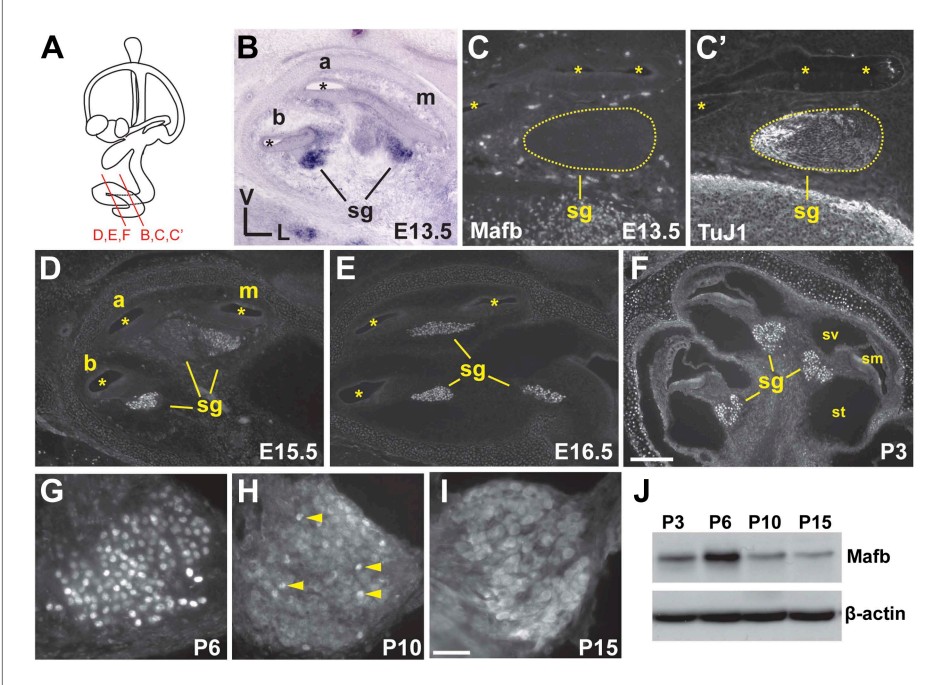

**Figure 1**. Mafb is expressed in SGNs during synaptogenesis. (**A**) Schematic representation of a mouse inner ear indicates the levels of section planes in (**B–F**). (**B**) In situ hybridization shows *Mafb* expression in E13.5 SGNs. (**C** and **C'**) Mafb (**C**) and Neuronal Class III β-Tubulin (TuJ1) (**C'**) double staining at E13.5 shows a lack of Mafb protein in the spiral ganglion (sg), which is marked by TuJ1. (**D–F**) Anti-Mafb immunostaining of the cochlea at E15.5 (**D**), E16.5 (**E**) and P3 (**F**) confirmed specific expression of Mafb in differentiating SGNs. Mafb protein can be detected in SGNs in the basal and middle turns at E15.5 (**D**) and in all SGNs at E16.5 (**E**). High expression persists through early postnatal stages (**F**). At all stages, Mafb protein is restricted to SGNs. Asterisks indicate the cochlear duct. a: apex; m: middle; b: base; sg: spiral ganglion; sv: scala vestibuli; sm: scala media; st: scala tympani. (**G–I**) Anti-Mafb immunostaining of transverse sections through the middle turn of the cochlea at P6 (**G**), P10 (**H**) and P15 (**I**). Mafb protein is present at high levels in SGN nuclei at P6, then gradually translocated to the cytoplasm. Only a few SGNs maintained nuclear expression of Mafb at P10 (arrowheads). Mafb is present at low levels in the cytoplasm of all SGNs at P15. (**J**) Western blots of P3, P6, P10, and P15 cochlear lysates using anti-Mafb and β-actin (loading control) antibodies. Mafb protein level peaks at P6 and decreases afterward. (12 cochleae from 6 mice per age group). Scale bar in **F** is 200 µm (**B**, **D–F**) or 100 µm (**C** and **C'**); and in **I** is 50 µm (**G–I**).

To uncover Mafb activities in the cochlea, we used the *Foxg1-Cre* driver (**Hebert and McConnell, 2000**) with the *Mafb^flox* allele and a previously reported *Mafb^GFP* null allele (**Moriguchi et al., 2006**). *Foxg1* is expressed in the entire otic vesicle as early as E8.5 (**Hebert and McConnell, 2000**), long before the onset of *Mafb* expression. Hence, in *Foxg1^Cre/+;Mafb^flox/GFP* mutants, Mafb is never produced in SGNs (**Figure 3A,B**). In contrast to previously described *Mafb* mutants (**Cordes and Barsh, 1994**; **Choo et al., 2006**), the inner ear developed normally in *Foxg1^Cre/+;Mafb^flox/GFP* mutants (**Figure 3C,D**). Thus, Mafb does not play a direct role in inner ear morphogenesis, consistent with previous conclusions that the inner ear phenotype is secondary to hindbrain malformation (**Choo et al., 2006**). The neurite organization of SGNs was also unaffected by the loss of Mafb (**Figure 3E,F**). Unfortunately, analysis of later functions was not possible as *Foxg1^Cre/+;Mafb^flox/GFP* mice died at birth, likely due to central apnea.

To examine Mafb function later in development, we generated a second conditional knock-out strain using the *Neurog1-Cre* driver (**Quinones et al., 2010**), which mediates recombination specifically in early SGNs (**Figure 3—figure supplement 1**). In *Tg(Neurog1-Cre);Mafb^flox/GFP* mice (hereafter referred to as *Mafb^CKO*), Mafb was significantly reduced at E16.5 (data not shown) and nearly undetectable by P5 (**Figure 3G–I**). However, unlike *Mafb* null mutants, *Mafb^CKO* mice live through adulthood, permitting analysis of auditory circuit assembly and function. As expected, early stages of SGN development proceeded normally in *Mafb^CKO* animals, with no obvious change in the organization of

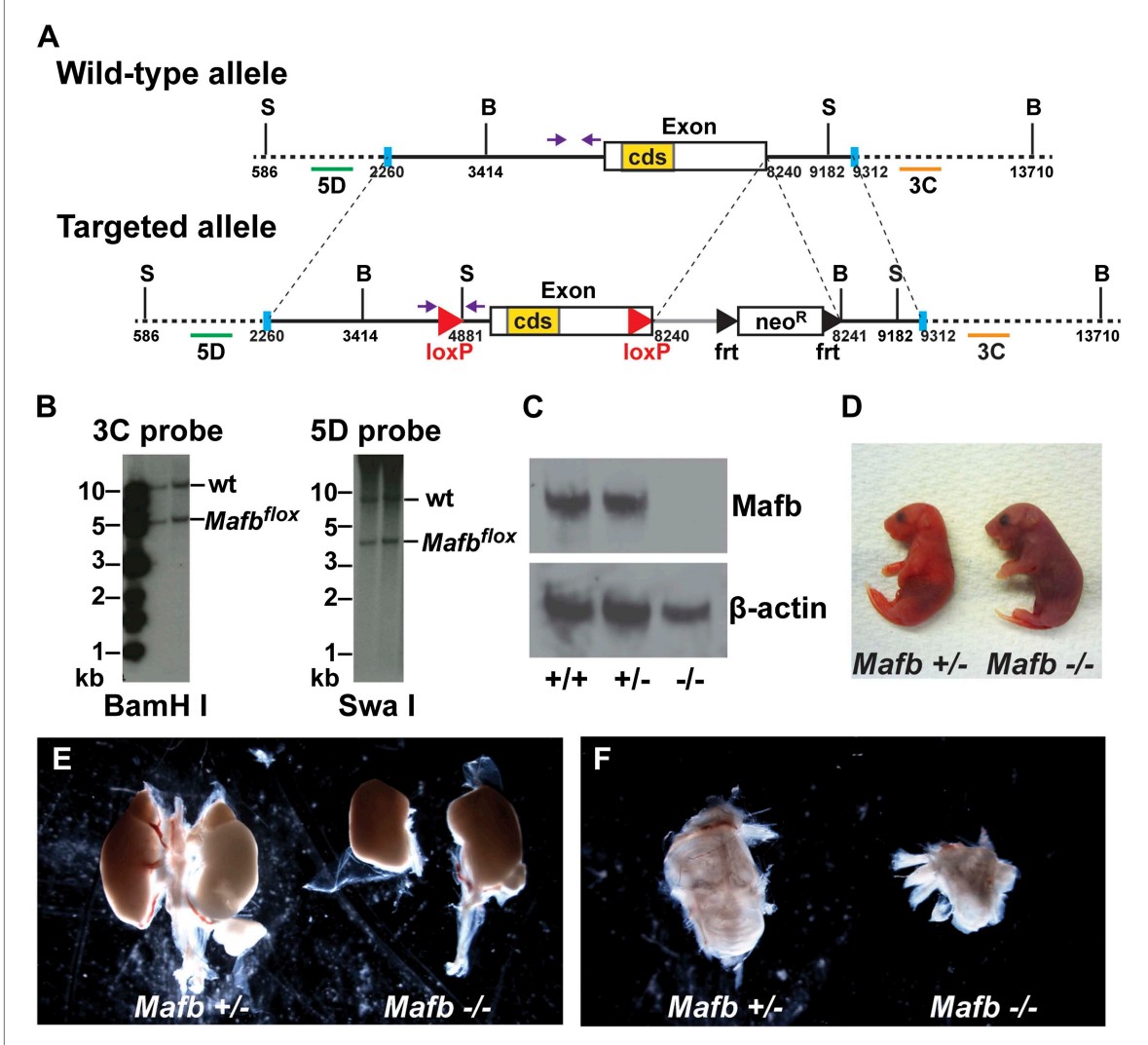

**Figure 2**. Generation and validation of floxed *Mafb* alleles. (**A**) A map of the *Mafb* locus in wild-type and floxed alleles. Two *LoxP* sites flank the Mafb coding region that is contained in a single exon. A *neoR* cassette was used for positive selection and can be removed by FLP-mediated recombination of *FRT* sites. B: BamHI site, S: SwaI site. cds: coding DNA sequence. Purple arrows indicate forward and reverse primers used for genotyping. Green and orange bars indicate 5D probes or 3C probes used for Southern blotting in (**B**). Blue vertical bars demarcate the sequence used to target to the *Mafb* genomic locus. (**B**) Southern blotting of the 3' (left) and 5' (right) *Mafb* genomic region confirmed successful homologous recombination in two different ES cell clones. (**C**) Western blot with anti-Mafb antibodies (top) confirms loss of protein from E16.5 *Mafb* null mutants generated from the *Mafb*^*flox* line. β-actin serves as a loading control. (**D**) *Mafb* null neonates take only occasional gasping breaths, show cyanosis, and die within 2 hours after birth. (**E**) E18.5 *Mafb* null mutants have dystrophic kidneys compared to control littermates. (**F**) *Mafb* null mutants develop a small and cystic inner ear compared to control littermates.

SGN processes (*Figure 3J,K*) or SGN density (*Figure 3L–N*). Taken together with results from the *Foxg1-Cre* driver, we conclude that Mafb is not required for the initial production or differentiation of SGNs.

## Mafb is essential for normal development of the auditory afferent synapse

The absence of abnormalities in early SGN development suggested that Mafb may play a critical role at later stages, perhaps during auditory synapse development. Ultrastructural analysis revealed that mutant SGNs still contact IHCs, indicating normal targeting (*Figure 4—figure supplement 1A,B*). We therefore asked whether these contacts were able to develop into synapses. Ribbon synapses were

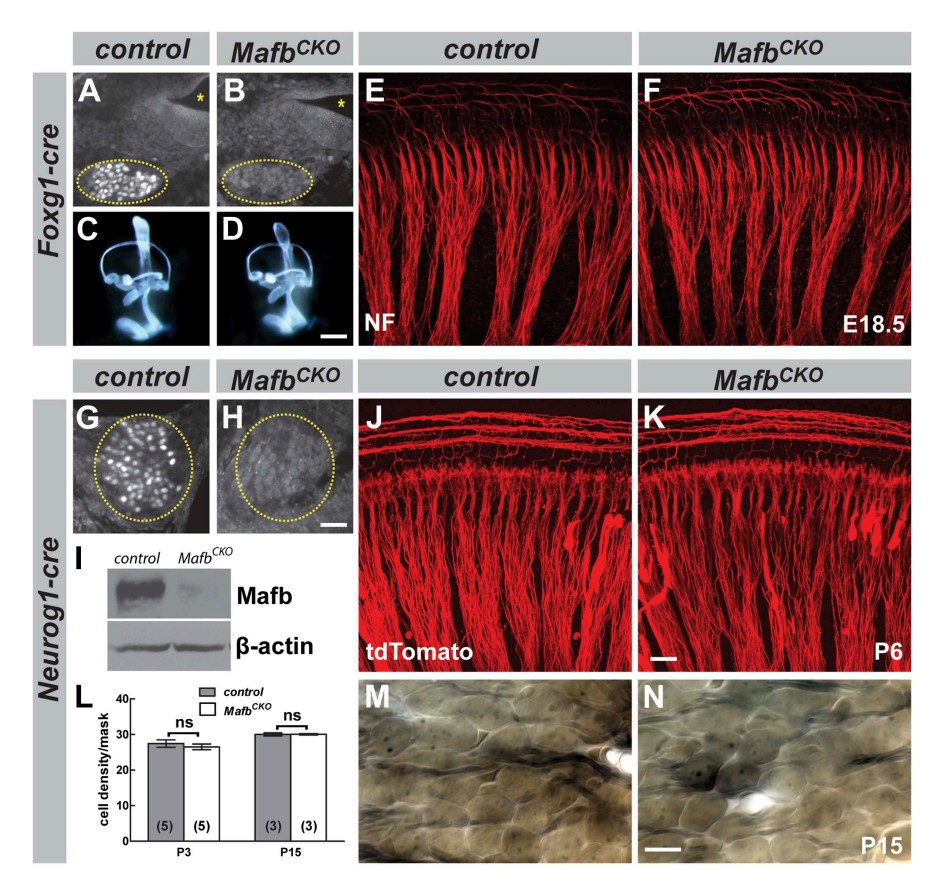

**Figure 3**. Mafb is not required for inner ear patterning or extension of SGN peripheral projections. (**A** and **B**) Anti-Mafb immunostaining of transverse sections through E16.5 control (**A**) and $Foxg1^{Cre/+};Mafb^{flox/GFP}$ mutant (**B**) cochleae confirms Mafb is not produced in mutant SGNs (circled area). (**C** and **D**) Light microscope images of E14.5 control (**C**) and $Foxg1^{Cre/+};Mafb^{flox/GFP}$ mutant (**D**) inner ears filled with paint show normal inner ear morphology in mutants. (**E** and **F**) Confocal stacks from E18.5 control (**E**) and $Foxg1^{Cre/+};Mafb^{flox/GFP}$ mutant (**F**) cochlear whole-mounts stained for Neurofilament (NF) to label all neuronal processes. Cochlear wiring is normal in mutant animals. (**G** and **H**) Anti-Mafb immunostaining of transverse sections through P5 control (**G**) and $Neurog1\text{-}Cre;Mafb^{flox/GFP}$ ($Mafb^{CKO}$) mutant (**H**) cochleae shows Mafb is not expressed in $Mafb^{CKO}$ SGNs (circled area). (**I**) Western blot using anti-Mafb and β-actin (loading control) antibodies on P5 cochlear lysate (20 cochleae from 10 mice per genotype) confirms loss of protein from $Mafb^{CKO}$ cochleae. (**J** and **K**) P6 cochlear whole-mounts from $Neurog1\text{-}Cre;Mafb^{flox/+};tdTomato$ (**J**) and $Neurog1\text{-}Cre;Mafb^{flox/GFP};tdTomato$ (**K**) mice imaged for tdTomato show afferent fibers from SGNs projecting normally in $Mafb^{CKO}$ cochlea. (**L**) SGN density was quantified in a masked area of the 32 kHz region of control and $Mafb^{CKO}$ cochleae at P3 and P15. Data are presented as average SGN cell density per mask for each group. SGN cell number is similar between control and $Mafb^{CKO}$ mice at P3 and P15. $p=0.51$ at P3 and 0.97 at P5. ns: not significant. In this and all subsequent figures, numbers in parentheses indicate the number of cochleae (one cochlea per mouse) used for quantification. (**M** and **N**) Representative light micrographs of osmium-stained plastic sections show similar cell densities of control and $Mafb^{CKO}$ cochleae at P15. Images corresponding to the masked region are shown. Scale bar in **D** is 100 μm (**A** and **B**) or 400 μm (**C** and **D**); in **H** is 50 μm (**G** and **H**); in **K** is 20 μm (**E**, **F**, **J**, **K**); and in **N** is 20 μm (**M** and **N**).

The following figure supplements are available for figure 3:

**Figure supplement 1**. *Neuorg1-Cre* drives Cre-mediated recombination in SGNs but not in olivocochlear efferent neurons.

assessed by staining for the postsynaptic marker GluR2, labeling a subunit of the AMPA receptor, and the presynaptic marker CtBP2, labeling the B domain of the RIBEYE scaffolding protein (**Schmitz et al., 2000**). The use of anti-GluR2 antibodies ensured that we only analyzed contacts between SGNs and IHCs, as other post-synaptic markers are also present in other types of synapses in the cochlea

(*Khimich et al., 2005*; *Nemzou et al., 2006*). Hence, ribbon synapses were reliably identified as juxtaposed pairs of immunofluorescent puncta of postsynaptic GluR2 and presynaptic CtBP2 (*Khimich et al., 2005*; *Liberman et al., 2011*). Because the number of IHC ribbon synapses differs along the tonotopic axis (*Meyer et al., 2009*), we were careful to compare the same region between cochleae (around 16 kHz, see 'Materials and Methods').

The analysis of GluR2 expression revealed an obvious failure in post-synaptic differentiation in *Mafb*^CKO animals. In control animals, diffuse patches of GluR2 immunofluorescence were present near the basolateral pole of the IHCs by P6 (*Figure 4A*, arrowhead). In contrast, GluR2 staining intensity was much lower in *Mafb*^CKO animals (*Figure 4B,K*) (*p*<0.03), indicating that GluR2 fails to accumulate postsynaptically. This loss of postsynaptic clustering of GluR2 appears to reflect an overall failure in PSD development, as ultrastructural studies revealed a clear disruption in PSD morphology at P6. During this period of active synapse development and refinement in the mouse cochlea, IHCs contain both anchored ribbons, which are localized to the cell surface, and floating ribbons, which are initially free in the cytoplasm and can become anchored as the cochlea matures. In control mice, anchored ribbons faced a large, well-defined PSD in the apposing SGN terminal (*Figure 4C,D*, *Figure 4—figure supplement 1E,F*). In contrast, *Mafb*^CKO ribbons were not paired with an obvious PSD even when in close contact with an afferent SGN terminal (*Figure 4E,F*, *Figure 4—figure supplement 1G,H*). Some ribbons appeared morphologically normal (*Figure 4E*), but others were abnormally small (*Figure 4F*) or misshapen (*Figure 4—figure supplement 1H*).

Closer examination confirmed that defective PSD differentiation secondarily affected presynaptic development. In P3 control cochleae, ribbons were dispersed broadly in the IHC cytoplasm, with many CtBP2-positive puncta not yet localized to the basolateral surface (*Figure 4—figure supplement 1C'*). Over the next several days, RIBEYE assembled into larger complexes that stained more brightly and were confined to the basal pole of the IHC (*Figure 4A'*). In contrast, anti-CtBP2 immunofluorescence intensity was noticeably reduced in *Mafb*^CKO mutants at P6 (*Figure 4B'*), consistent with observation of small ribbons in electron micrographs (*Figure 4F*). In addition, ribbon number was significantly decreased (*Figure 4L*, *Table 1*). Although GluR2 immunofluorescence was not bright enough to assess PSD differentiation before P6 (*Figure 4—figure supplement 1C,D*), CtBP2 intensity was also diminished at P3 (*Figure 4—figure supplement 1D'*), suggesting that SGN post-synaptic differentiation is impaired from the earliest stages. Thus, *Mafb* mutant SGNs fail to elaborate the specialized PSDs that are characteristic of auditory ribbon synapses, apparently leading to defects in the development and anchoring of presynaptic ribbons in the IHCs.

These reciprocal defects in early pre- and post-synaptic development resulted in a severe disruption in the mature pattern of synaptic connectivity, with an ~50% reduction in the number of GluR2 puncta and an ~30% reduction in ribbon number at P15 (*Figure 4G–H',K,L*, *Table 1*) and in adults (*Figure 4K,L*, *Figure 4—figure supplement 1I-J'*, *Table 1*). The loss of GluR2 and CtBP2 puncta was observed along the entire length of the mutant cochlea (data not shown). Additionally, close examination of individual pre- and post-synaptic puncta showed that many CtBP2-positive ribbons were not paired with a corresponding GluR2 spot (*Figure 4I,J*). This stands in contrast to control animals, where an individual ribbon is reliably juxtaposed to a single GluR2-positive terminal by this stage (*Khimich et al., 2005*; *Nemzou et al., 2006*; *Sendin et al., 2007*). Hence, the overall loss of synapses seems to originate with a defect in the PSD. This phenotype is not due to a general decrease in the transcription of PSD components, as the expression levels of *GluR2* and *PSD95* transcripts were similar between control and *Mafb* mutant SGNs (*Figure 4—figure supplement 2*).

Although olivocochlear efferent neurons also express Mafb (NR Druckenbrod and LVG, unpublished observation) and are thought to influence cochlear development (*Guinan, 2010*), the loss of afferent synapses is unlikely to be secondary to abnormalities in the efferent system since *Neurog1-Cre* is not active in olivocochlear neurons (*Figure 3—figure supplement 1*). In addition, the efferent innervation of the *Mafb*^CKO cochlea is normal (*Figure 4—figure supplement 3*). Thus, we conclude that Mafb acts in SGN afferents to specify post-synaptic differentiation of ribbon synapses.

## Mafb is required for normal auditory function

Ribbon synapses transmit all acoustic information from IHCs to SGNs, playing a critical role in shaping the final perception of sound. To determine how auditory function is altered when synapse development is disrupted, we measured Auditory Brainstem Responses (ABRs) in *Mafb*^CKO animals. ABRs reflect the electrical responses of neurons in the cochlea and auditory brainstem to sound stimuli. Acoustic

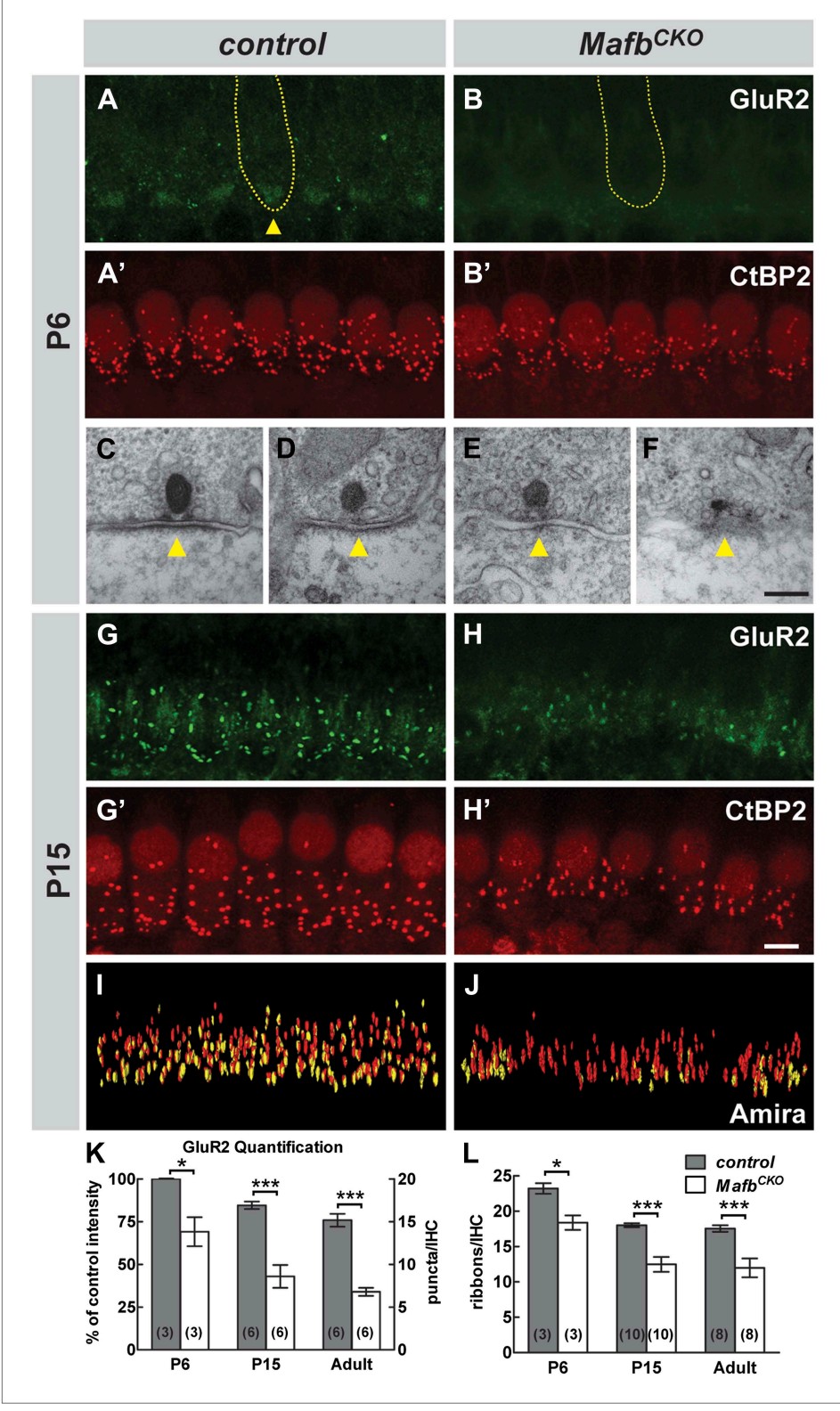

**Figure 4**. Loss of Mafb interferes with PSD development and reduces synapse number. (**A–B′**) Confocal stacks from P6 control (**A** and **A′**) and *Mafb^CKO* (**B** and **B′**) cochlear whole-mounts double stained for GluR2 (green) and CtBP2 (red). IHC nuclei are also weakly immunopositive for CtBP2. Yellow dotted lines (**A** and **B**) outline IHCs, as determined by Myo7A staining in another channel (not shown). GluR2 immunofluorescence is present near the basal
*Figure 4. Continued on next page*

*Figure 4. Continued*

pole of control (arrowhead) but not *Mafb^CKO* IHCs. The intensity of CtBP2 immunofluorescence is decreased in *Mafb^CKO* IHCs at P6 when compared to controls. (**C–F**) Electron micrographs of IHC ribbon synapses in the middle turn of cochleae from P6 control (**C** and **D**) and *Mafb^CKO* (**E** and **F**) mice. *Mafb^CKO* mice lack a delineated PSD in their SGN afferent terminals (arrowheads). (**G–H′**) Confocal stacks from P15 control (**G** and **G′**) and *Mafb^CKO* (**H** and **H′**) cochlear whole-mounts double stained for GluR2 (green) and CtBP2 (red). The number of GluR2 and CtBP2 puncta is decreased in P15 *Mafb^CKO* mice. (**I** and **J**) 3D reconstructions of presynaptic ribbons (red) and postsynaptic aggregations of GluR2 (yellow) from P15 control (**I**) and *Mafb^CKO* (**J**) cochleae derived from confocal stacks using Amira image-processing software. Many *Mafb^CKO* ribbons lack corresponding GluR2 puncta (**J**). (**K**) Quantification of GluR2 immunofluorescence intensity at P6 (left axis) and GluR2 puncta number per IHC at P15 and adult (right axis) in control and *Mafb^CKO* cochleae. GluR2 immunofluorescence intensity of P6 *Mafb^CKO* cochleae is expressed as a percentage of control intensity. (**L**) Quantification of ribbon number per IHC in control and *Mafb^CKO* cochleae at P6, P15 and adult. Ribbon number and GluR2 intensity/number of *Mafb^CKO* cochleae are significantly decreased when compared to controls at all stages. *: $p<0.05$, ***: $p \leq 0.001$. Scale bar in **H′** is 5 µm (**A**, **A′**, **B**, **B′**, **G**, **G′**, **H**, **H′**); in **F** is 200 nm (**C–F**).

The following figure supplements are available for figure 4:

**Figure supplement 1**. *Mafb^CKO* mutants show normal targeting of afferent terminals but impaired synapse development.

**Figure supplement 2**. Transcription of *GluR2* and *PSD95* is not significantly changed in *Mafb^CKO* SGNs.

**Figure supplement 3**. The olivocochlear efferent system is normal in *Mafb^CKO* mice.

sensitivity is determined by identifying the lowest sound pressure level (SPL) (the threshold) that is able to elicit a response. ABRs were recorded in response to pure tone bursts of 5.6, 8, 11.3, 16, 22.6, 32, and 45.2 kHz. Control mice produced characteristic ABR waveforms in response to all stimuli, with thresholds ranging between 15 and 45 dB SPL depending on the frequency of the stimulus. In contrast, *Mafb^CKO* mice showed dampened ABR waveforms and elevated auditory thresholds, with thresholds increased ~10 dB SPL across stimuli of all frequencies (***Figure 5A,B,D***, ***Figure 5—figure supplement 1***).

**Table 1.** Quantitative analysis of IHC ribbon synapses

| | | GluR2 puncta number per IHC (mouse number) | *p*-value | CtBP2 puncta number per IHC (mouse number) | *p*-value |
|---|---|---|---|---|---|
| P6 | *CE control* | n/a | n/a | 23.0 ± 0.6 (6) | 0.0750 (ns) |
| | *Mafb^CE* | n/a | | 24.5 ± 0.4 (6) | |
| | *CKO control* | n/a | n/a | 23.2 ± 0.7 (3) | 0.0187 |
| | *Mafb^CKO* | n/a | | 18.4 ± 1.0 (3) | |
| P15 | *CE control* | 16.6 ± 0.9 (3) | 0.6916 (ns) | 17.9 ± 0.5 (3) | 0.3757 (ns) |
| | *Mafb^CE* | 17.1 ± 0.2 (3) | | 18.6 ± 0.4 (3) | |
| | *CKO control* | 16.9 ± 0.3 (6) | 0.0001 | 18.0 ± 0.3 (10) | $7.6 \times 10^{-5}$ |
| | *Mafb^CKO* | 8.6 ± 1.0 (6) | | 12.5 ± 1.0 (10) | |
| | *Gata3^CKO control* | 15.3 ± 0.4 (7) | $6.4 \times 10^{-9}$ | 17.1 ± 0.6 (7) | $1.4 \times 10^{-7}$ |
| | *Gata3^CKO* | 5.6 ± 0.5 (7) | 0.0003 | 8.2 ± 0.5 (7) | 0.0001 |
| | *Gata3^CKO;Mafb^CE* | 10.0 ± 0.7 (7) | | 12.3 ± 0.6 (7) | |
| Adult | *CKO control* | 15.2 ± 0.7 (6) | $2.7 \times 10^{-6}$ | 17.6 ± 0.5 (8) | 0.0014 |
| | *Mafb^CKO* | 6.8 ± 0.4 (6) | | 12.0 ± 1.3 (8) | |

Means ± SEMs are shown. *p*-values were obtained by Student's *t*-test. Quantification and statistical results of GluR2 and CtBP2 puncta from all mouse strains used in this study.

Thresholds of $Mafb^{CKO}$ mice were significantly higher for all frequencies measured except the lowest and highest frequencies, since sensitivity is already reduced in these regions of control cochleae (*Figure 5B*) ($p$ values: 5.6 kHz=0.125, 8 kHz=0.044, 11.3 kHz=0.043, 16 kHz=0.031, 22.6 kHz=0.038, 32 kHz=0.047, 45.2 kHz=0.140; n = 16 controls and 16 mutants). We also investigated whether this impaired auditory response involved an OHC defect by recording distortion product otoacoustic emissions (DPOAEs). DPOAE levels were not significantly different in $Mafb^{CKO}$ vs control mice, regardless of the frequency and intensity of the stimulus used (*Figure 5C*), indicating that hair cell function is largely normal. Hence, the ABR threshold elevation is mainly caused by the reduced activation of SGNs by sound stimuli.

We further characterized the change in SGN responsiveness by analyzing Wave I of the ABR waveforms, which represents the summed activity of SGNs evoked by sound. The strength and speed of the electrical response were evaluated by measuring the amplitude and latency of wave I. In $Mafb^{CKO}$ mice, Wave I was delayed, with increased latency in mutants exposed to a 16 kHz pure tone (*Figure 5E*). More strikingly, the amplitude was significantly diminished in response to stimuli across frequencies and intensities (*Figure 5D,F,G*, *Figure 5—figure supplement 1*). For instance, the response to a 16 kHz, 80 dB stimulus was 35% smaller in mutants than in controls. Hence, loss of Mafb prevents the development of normal acoustic responsiveness, with effects along the length of the cochlea.

## $Mafb^{CKO}$ SGNs have normal firing properties

ABR responses reflect the overall activation of SGNs, which is determined both by the strength of the synaptic signals and by the intrinsic response properties of the SGNs. We wondered whether the diminished ABR responses in mutants could be explained entirely by the loss of synapses or whether other aspects of SGN responsiveness were also altered. Electrophysiological recordings offer a highly sensitive read-out of SGN function that can reveal differences in membrane potential and firing properties. Accordingly, we performed whole-cell recordings of SGNs in organotypic explants, acutely dissected at P3. Analysis at P3 allows us to focus on SGN properties independent of any secondary effects from loss of synaptic transmission in older animals.

SGNs express several voltage-dependent ionic currents that contribute to their membrane properties (*Santos-Sacchi, 1993*; *Adamson et al., 2002*). We recorded voltage-dependent currents in voltage-clamp mode to assess membrane properties of SGNs. Voltage steps evoked prominent inward and outward currents, indicative of voltage-dependent sodium and potassium currents (*Figure 6A*). Maximum peak outward current amplitudes were not significantly different between $Mafb^{CKO}$ mutants and littermate controls ($Mafb^{CKO}$: 4.1 ± 1.3 nA, n = 11; control: 4.4 ± 0.8 nA, n = 10) and there was no significant difference in the current–voltage relationships (*Figure 6B*, lower left panel). We also analyzed fast activating and fast inactivating inward currents, characteristic of voltage-dependent sodium currents ($I_{Na}$) (*Figure 6A*). Again, no significant difference was found in $I_{Na}$ (*Figure 6B*, lower right panel) ($Mafb^{CKO}$:−1.2 ± 1.5 nA, n = 7; controls: −2.1 ± 0.9 nA, n = 5).

To assess SGN firing properties, we recorded from $Mafb^{CKO}$ SGNs in current-clamp mode. Depolarizing current injections above 30 pA evoked action potentials (APs) in $Mafb^{CKO}$ SGNs (*Figure 6C*), suggesting that the AP firing machinery was not compromised. Quantitative analyses of AP parameters, such as resting membrane potential, AP threshold, and AP latency, indicated no significant difference between $Mafb^{CKO}$ and control littermates (*Figure 6D*).

In summary, our results indicate that intrinsic membrane and firing properties were unaltered in postnatal $Mafb^{CKO}$ SGNs. We therefore propose that the elevated ABR threshold and reduced eighth nerve synchronization (*Figure 5*) is a direct consequence of the IHC-SGN synapse defect (*Figure 4*). Moreover, the emergence of normal SGN firing properties seems to be independent of *Mafb*.

## Mafb overexpression accelerates afferent synapse development

The absence of electrophysiological defects in mutant SGNs indicated that Mafb may play an unusually specific role directing postsynaptic differentiation. Consistent with this idea, in wild-type mice, Mafb protein is first detected just prior to the beginning of synapse development (*Figure 1*), as if the onset of Mafb expression triggers formation of the PSD in SGN terminals. To test this idea, we created a strain of *Mafb* conditional expressor mice ($Mafb^{CE}$) that produce exogenous Mafb upon *Neurog1-Cre*-mediated recombination [*Tg(Neurog1-Cre);Gt(ROSA)26Sor$^{CAG-lsl-Mafb}$*] (*Figure 7A*). In these mice, Mafb protein was present in SGNs by E12.5, at least 3 days earlier than the expression of endogenous

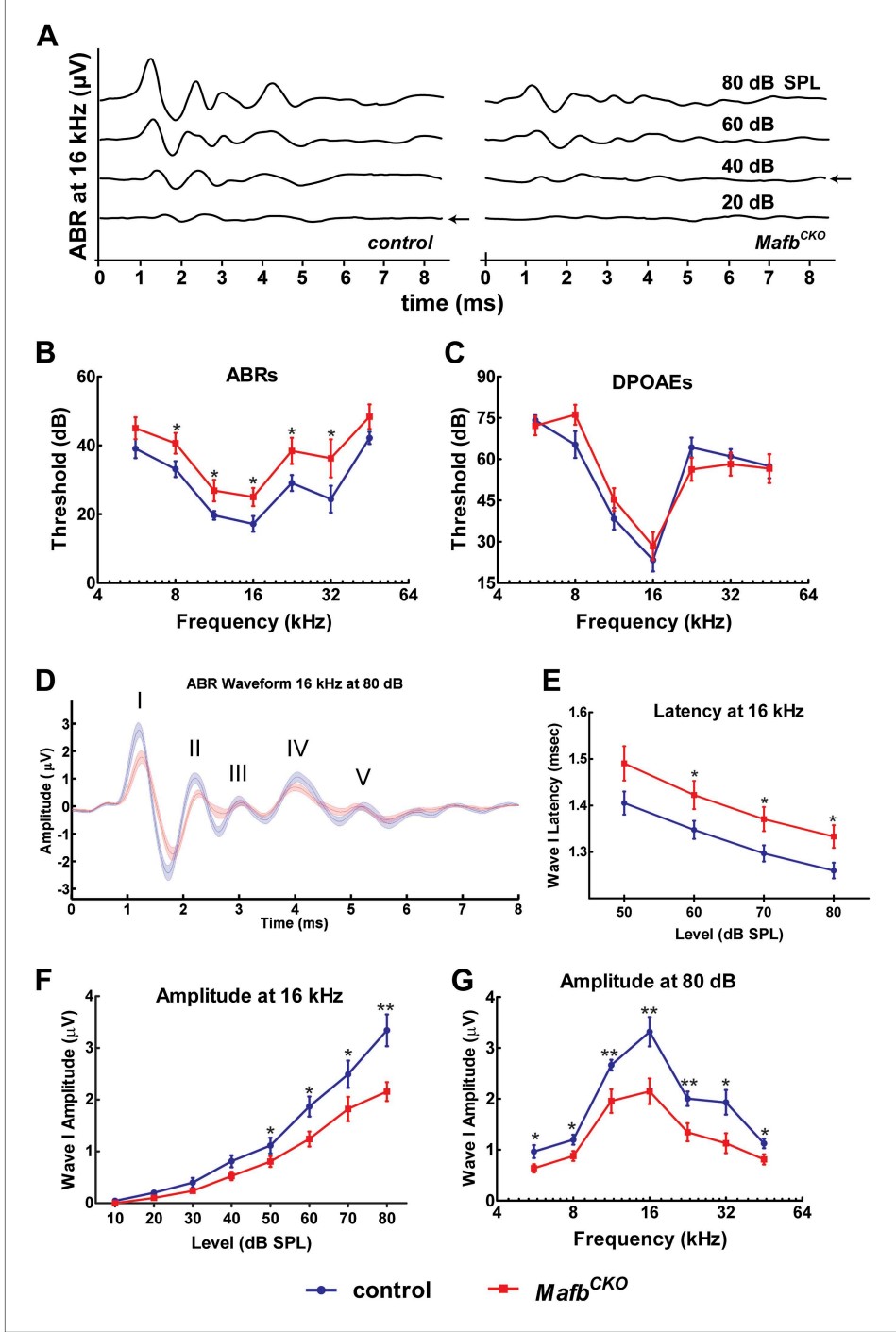

**Figure 5**. Mafb is required for normal auditory function. (**A**) Representative ABR recordings from a P41 control and *Mafb^CKO* littermate exposed to a 16 kHz pure tone stimulus at intensities ranging from 20 to 80 dB. Arrows indicate the ABR threshold. The ABR response is diminished in the *Mafb^CKO* mutant compared to its littermate control. (**B** and **C**) Plots of threshold values from recordings of ABRs (**B**) and DPOAEs (**C**) performed on 5- to 7-week-old control (blue line) and *Mafb^CKO* (red line) mice. Auditory responses were assessed at 7 frequencies (5.6 kHz, 8 kHz, 11.3 kHz, 16 kHz, 22.6 kHz, 32 kHz, and 45.2 kHz), and across a range of sound pressure levels from 10 to 80 decibels. *: $p<0.05$. n = 16 mice for each group. *Mafb^CKO* mice show elevated ABR but normal DPOAE thresholds when compared to controls. (**D**) 16 kHz ABR waveforms of 16 control (blue line) and 16 *Mafb^CKO* (red line) mice were averaged and overlaid. Roman numerals mark the peaks of the ABR waves. Wave I is delayed and diminished in *Figure 5. Continued on next page*

*Figure 5. Continued*

*Mafb*<sup>CKO</sup> mutants. (**E**) Average Wave I latencies for control (blue, n = 16) and *Mafb*<sup>CKO</sup> (red, n = 16) littermates in response to a 16 kHz pure tone stimulus. *Mafb*<sup>CKO</sup> responses are significantly delayed at 80, 70, and 60 dB SPL. *: *p*<0.05. (**F**) Average Wave I amplitudes for control (blue, n = 16) and *Mafb*<sup>CKO</sup> (red, n = 16) littermates in response to a 16 kHz stimulus. The *Mafb*<sup>CKO</sup> responses are significantly decreased at 80, 70, 60, and 50 dB SPL. **: *p*<0.005 and *: *p*<0.05. (**G**) Average Wave I amplitudes for control (blue, n = 16) and *Mafb*<sup>CKO</sup> (red, n = 16) littermates at all 7 frequencies of the stimulus. Stimuli were presented at a sound pressure level of 80 dB. *Mafb*<sup>CKO</sup> responses are significantly decreased at all frequencies. **: *p*<0.01 and *: *p*<0.05.
The following figure supplements are available for figure 5:

**Figure supplement 1**. *Mafb*<sup>CKO</sup> mutants show diminished Wave I of ABRs in response to all frequencies.

---

Mafb protein in controls (*Figure 7B–C'*). Overexpression of Mafb had no gross effect on cochlear wiring (*Figure 7—figure supplement 1A,B*). However, AMPA receptors clustered earlier in SGN terminals, with multiple patches of GluR2 staining covering the entire basal pole of the IHCs (*Figure 7D,E*). Quantification confirmed a dramatic increase of GluR2 staining intensity in *Mafb*<sup>CE</sup> afferent terminals (*Figure 7F*) (*p*=0.033), with similar effects observed for a second PSD marker, PSD95 (*Figure 7—figure supplement 1C,D*). Hence, increased expression of Mafb enhanced post-synaptic differentiation in SGNs.

Since abnormal development of the PSD was accompanied by a change in presynaptic differentiation in *Mafb*<sup>CKO</sup> mice, we asked whether these changes in the development of the PSD would have any effect on the maturation of ribbons in IHCs. At P6, ribbons are gradually relocating from the cytoplasm to the membrane, where individual ribbons are anchored opposite an afferent terminal. In *Mafb*<sup>CE</sup> mice, this process occurred earlier and CtBP2-positive spots were noticeably more confined to the basal pole of the IHCs (*Figure 7D',E'*). To assess the distribution of ribbons independent of the angle at which each IHC was imaged, we measured the distance from each ribbon to its nearest neighbor; this distance is predicted to be shorter when ribbons are more tightly localized (*Figure 7H*). Indeed, the nearest-neighbor distance of ribbons in P6 *Mafb*<sup>CE</sup> mice was significantly shorter (1.83 ± 0.03 μm, n = 428 ribbons) than in controls (2.06 ± 0.04 μm, n = 407 ribbons) (*p*<0.0001) (*Figure 7I*). Hence, overexpression of Mafb caused synapses to develop earlier, as evidenced by increased accumulation of PSD components in SGN terminals and precocious basolateral localization of pre-synaptic ribbons in the hair cells.

Notably, this acceleration in synapse development did not cause any increase in the number of ribbons (*Figure 7D',E',G*, *Figure 7—figure supplement 1E',F'*, and *Table 1*) or GluR2-positive puncta (*Figure 7F*, *Figure 7—figure supplement 1E,F*, and *Table 1*) at P15. Ribbon distribution was also unchanged in P15 *Mafb*<sup>CE</sup> mice compared to controls (*p*=0.63) (*Figure 7I*). Thus, the pruning and refinement of synapses appear to be controlled by an independent pathway that is unaffected by the overexpression of Mafb.

Electrophysiological studies further confirmed that early and increased expression of Mafb does not seem to have adverse effects on SGN maturation. *Mafb*<sup>CE</sup> SGNs exhibited normal firing properties (*Figure 6*), with no change in the peak outward current amplitudes (*Mafb*<sup>CE</sup>: 4.8 ± 1.2 nA n = 5; control: 5.7 ± 1.0 nA, n = 6) (*Figure 6B*, upper left panel), current–voltage relationships, or inward currents (*Mafb*<sup>CE</sup>: −1.6 ± 0.5 nA, n = 5; control: −1.8 ± 0.4 n = 6) (*Figure 6B*, upper right panel). All AP parameters were also normal in *Mafb*<sup>CE</sup> SGNs (*Figure 6C,D*). Unfortunately, the majority (>80%) of *Mafb*<sup>CE</sup> mice die at ~3 weeks of age due to unknown causes, so it was not possible to test whether hearing develops normally in these animals. Nevertheless, our findings suggest that expression of Mafb is sufficient to initiate synapse development, without disrupting synaptic pruning or other aspects of SGN function.

## Mafb acts downstream of Gata3 to control SGN differentiation

Our studies show that Mafb promotes a key feature of the SGN identity, namely the formation of the uniquely large post-synaptic terminals that are necessary for the sense of hearing. This represents a late event in the execution of neuronal identity programs that are set in motion and coordinated by the early acting transcription factor Gata3 (*Appler et al., 2013*). One of Gata3's function appears to be the activation of additional SGN-specific transcription factors, including Mafb. In the immune

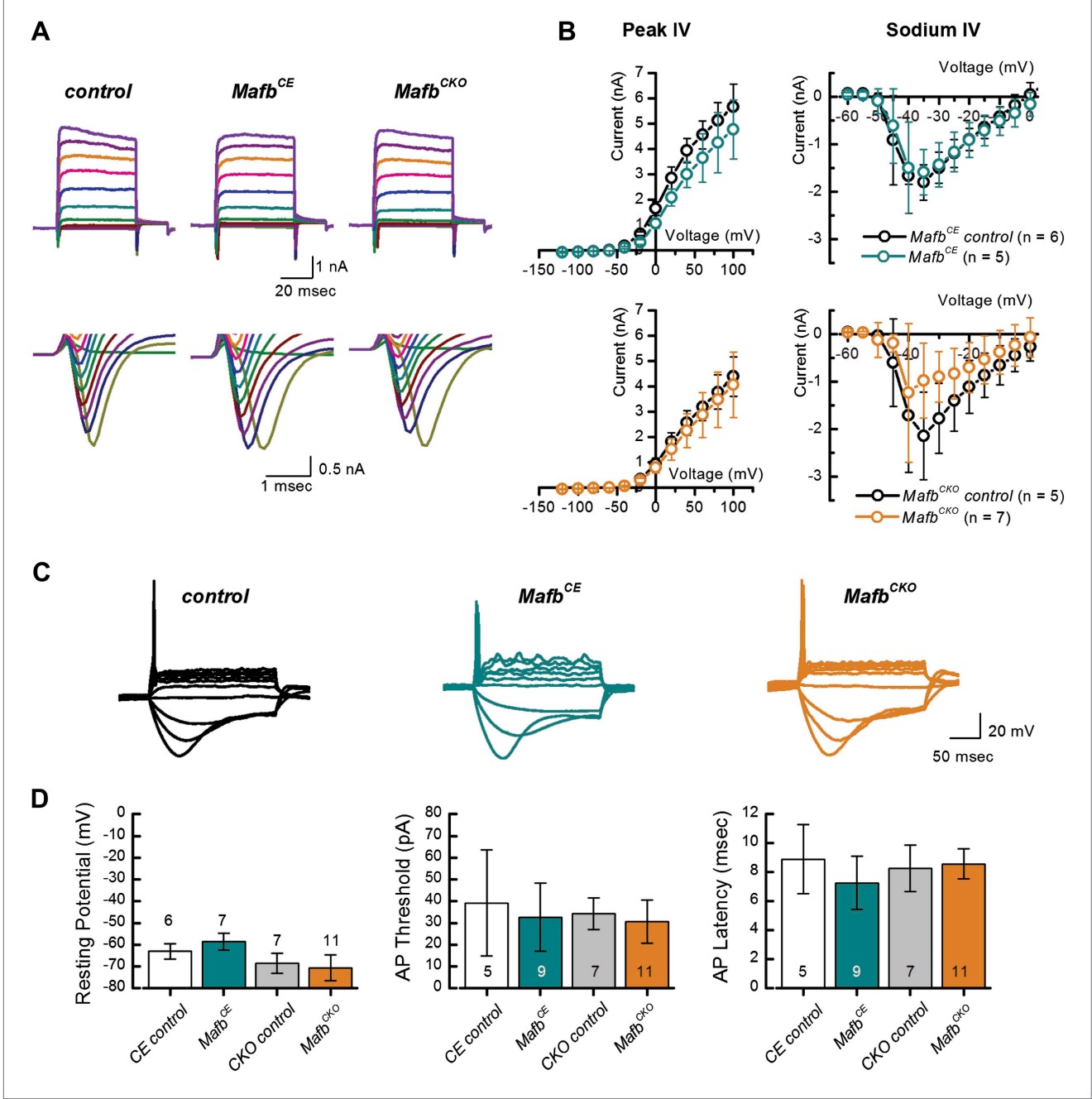

**Figure 6**. Mafb is not required for development of SGN firing properties. (**A**) Representative families of voltage-dependent outward K⁺ currents (top) and Na⁺ currents (bottom) recorded from SGNs in organotypic explants harvested from control, *Mafb^CE, and Mafb^CKO* animals at P3. K⁺ currents were evoked by voltage steps from −124 mV to 104 mV in 10 mV increments from a holding potential of −84 mV. Na⁺ currents were evoked by 100 ms prepulse to −104 mV from a holding potential of −84 mV, followed by voltage steps from −64 mV to 24 mV in 10 mV increments. Some current traces were removed for clarity. (**B**) Mean (±1 SD) potassium current–voltage (I–V) (left) and mean (±1 SD) sodium I–V relations (right). Number of SGNs for each genotype is indicated in the legend. (**C**) Representative current-clamp data recorded from P3 control, *Mafb^CE, and Mafb^CKO* SGNs in response to 200-ms current injections in 10 pA increments from rest. Depolarizing current injections elicit action potentials in littermate control, *Mafb^CE, and Mafb^CKO* SGNs and all exhibit prominent voltage sag in response to hyperpolarizing current injections. (**D**) Summary graphs of resting membrane potential (left), action potential (AP) threshold (middle), and AP latency (right) measured from Mafb^CE (dark cyan bar) *Mafb^CKO* (orange bar) and their littermate controls, denoted as CE control (white bar) and CKO control (gray bar) in the graphs, respectively. Resting membrane potential was measured as membrane voltage at I = 0 (no current injection). AP threshold was measured as the minimum current required to evoke an AP. AP latency was quantified as the time from the current step to the AP peak for a 50-pA current step.

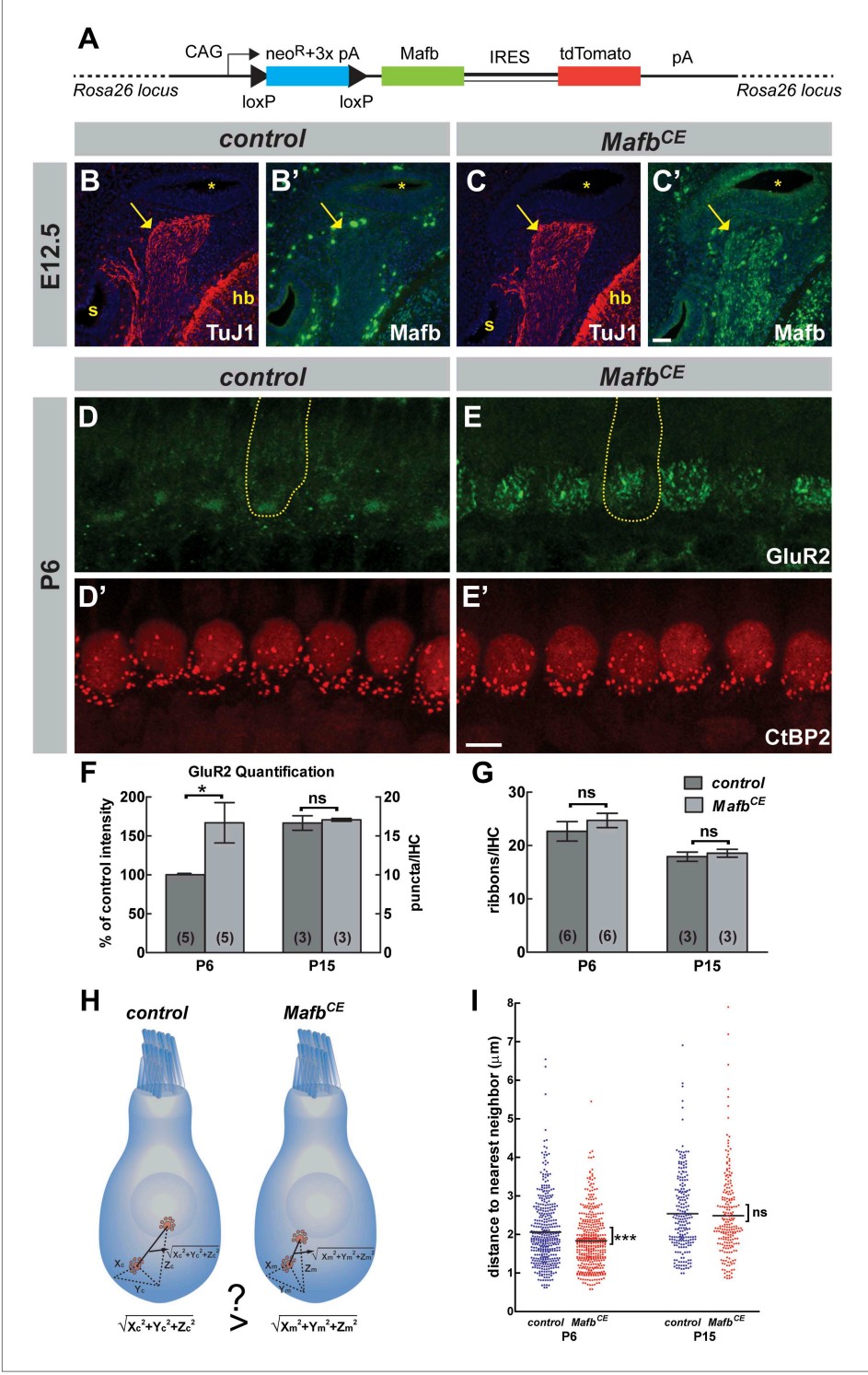

**Figure 7**. Overexpression of Mafb accelerates synapse development. (**A**) *Mafb^CE* mice have a Cre-dependent CAG promoter-driven *Mafb-ires-tdTomato-pA* cassette inserted into the *Rosa26* locus. (**B–C'**) Mafb (green) and TuJ1 (red, to mark spiral ganglia indicated by arrows) immunostaining on transverse sections of E12.5 control (**B** and **B'**) and *Mafb^CE* (**C** and **C'**) heads. The cochlear duct (asterisks), hindbrain (hb) and saccule (s) are visualized with DAPI counterstain (blue). Mafb is present in E12.5 *Mafb^CE* SGNs but not control SGNs. (**D–E'**) P6 control (**D** and **D'**) and *Mafb^CE* (**E** and **E'**) cochleae double stained for GluR2 (green) and CtBP2 (red). Yellow dotted lines (**D** and **E**) outline IHCs, as determined by Myo7A staining in another channel (not shown). GluR2 immunostaining shows multiple

*Figure 7. Continued on next page*

*Figure 7. Continued*

intense fluorescence patches in the basal region of *Mafb^CE* IHCs compared to a single patch in controls. (**F**) Quantification of P6 GluR2 immunofluorescence intensity (left axis) and P15 GluR2 puncta number per IHC (right axis) in control and *Mafb^CE* cochleae. GluR2 immunofluorescence intensity of P6 *Mafb^CE* cochleae is expressed as a percentage of control intensity. GluR2 intensity of P6 *Mafb^CE* cochleae is significantly increased compared to controls. P15 GluR2 puncta number is similar between control and *Mafb^CE* mice. (**G**) Quantification shows similar ribbon number per IHC in control and *Mafb^CE* cochlea at P6 and P15. *: $p<0.05$. ns: not significant. (**H**) Illustration of how the 3D distance between two ribbons was measured. (**I**) Scatter plot of the distance to nearest neighbor of P6 and P15 control (blue) and *Mafb^CE* (red) ribbons. Middle bar represents mean. ***: $p<0.001$. ns: not significant. n=407 control and 428 *Mafb^CE* ribbons in 18 IHCs from 6 mice at P6, 217 control and 215 *Mafb^CE* ribbons in 12 IHCs from three mice at P15. P6 *Mafb^CE* ribbons are significantly closer to their nearest neighbors compared to controls, reflecting an overall increased confinement of ribbons to the basal pole of the IHC. Scale bar in **C'** is 50 μm (**B–C'**); in **E'** is 5 μm (**D–E'**).

The following figure supplements are available for figure 7:

**Figure supplement 1**. *Mafb^CE* mice show precocious development of IHC ribbon synapses.

---

system, Gata3 cooperates with the large Maf factor c-Maf to control the terminal differentiation of Th2 helper cells (*Ho et al., 1998*). We therefore hypothesized that Mafb similarly mediates Gata3's effects on SGN-specific features of development.

We tested this hypothesis by investigating *Mafb* expression and function in *Gata3* conditional knock-outs (*Gata3^CKO*). Loss of Gata3 from SGNs using the *Bhlhe22-Cre* driver severely disrupts cochlear innervation (*Appler et al., 2013*). In *Gata3^CKO* mutants, Mafb protein levels were strongly reduced (*Figure 8B'*). Consistent with this loss of Mafb, the number of GluR2 and CtBP2 puncta was significantly decreased in P15 *Gata3^CKO* mutants (*Figure 8E,E',G* and **Table 1**). Synaptic loss in *Gata3^CKO* (*Figure 8G*) was more severe than what occurs in *Mafb^CKO* animals (*Figure 4K,L*), which fits with Gata3's earlier and broader effects on SGN neurite outgrowth, axon guidance, and survival (*Appler et al., 2013*; *Duncan and Fritzsch, 2013*; *Luo et al., 2013*). This makes it difficult to conclude whether the synaptic loss is due to the loss of *Mafb* or to other aspects of the *Gata3* mutant phenotype.

To clarify the relationship between *Gata3* and *Mafb* in SGNs, we asked whether restoration of *Mafb* is sufficient to rescue synapse development in the *Gata3^CKO* cochlea. We used the *Mafb^CE* line to provide exogenous Mafb into the *Gata3^CKO* background. In *Bhlhe22^Cre/+;Gata3^flox/tauLacZ;ROSA26^CAG-lsl-Mafb* animals (i.e., *Gata3^CKO;Mafb^CE*), Mafb protein was present in SGN nuclei, despite the absence of Gata3 (*Figure 8C,C'*). Strikingly, this restoration of Mafb was sufficient to improve the synaptic defect seen in *Gata3^CKO* mutants (compare *Figure 8F,F'* to *Figure 8E,E'*), with an ~80% increase in GluR2 puncta and an ~50% increase in ribbon number in *Gata3^CKO;Mafb^CE* mice when compared to *Gata3^CKO* littermates (*Figure 8G*, **Table 1**). Synapses could also be rescued by resupplying Mafb to *Gata3^CKO* mice made using the *Neurog1-Cre* line (*Figure 8—figure supplement 1*), indicating that the effect is SGN-autonomous. Taken together, these findings indicate that Gata3 and Mafb are key players in a transcriptional cascade that guides SGN development and ensures the emergence of the cell-type specific features that are critical for the sense of hearing.

## Discussion

In this study, we showed that the emergence of cell-type specific features of synapse development can be controlled by late acting lineage-specific transcription factors, with Mafb acting downstream of Gata3 to ensure formation of the specialized ribbon synapses that mediate the sense of hearing. Our results indicate that Mafb plays a highly specific role, with dramatic effects on post-synaptic differentiation in the absence of any other obvious changes in SGN function. Indeed, Mafb was capable of restoring synapses even in the *Gata3* mutant background, where SGN development is severely disrupted. These findings establish Mafb as a potent regulator of SGN terminal differentiation that functions within a broader Gata3 transcriptional network, offering a potent molecular entry point for designing new treatments for age-related and noise-induced hearing loss.

Efforts to elucidate the molecular basis of neuronal differentiation have highlighted the importance of transcriptional networks that progressively specify neuronal fates and direct the proper execution of these fates during terminal differentiation (*Hobert, 2011*). The earliest acting transcription factors

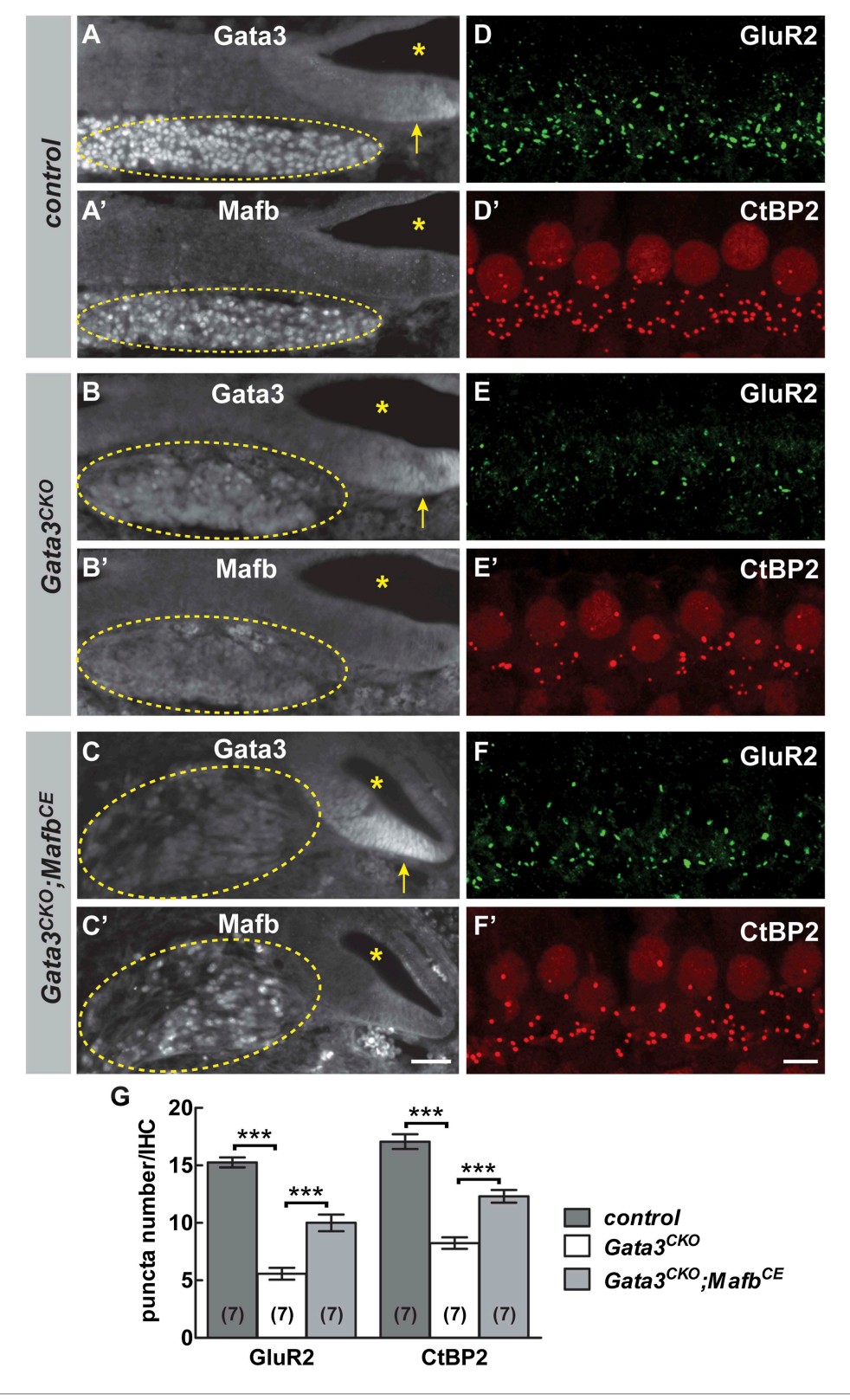

**Figure 8**. Mafb acts downstream of Gata3 to control synapse development. (**A-C'**) Transverse sections of E16.5 control (**A** and **A'**), *Gata3*^CKO (**B** and **B'**) and *Gata3*^CKO;*Mafb*^CE (**C** and **C'**) cochleae double immunostained with antibodies against Gata3 (**A**, **B**, **C**) and Mafb (**A'**, **B'**, **C'**). Asterisks indicate the cochlear duct. Gata3 and Mafb are
*Figure 8. Continued on next page*

*Figure 8. Continued*

normally co-expressed in SGNs at E16.5 (**A** and **A'**). In *Gata3^CKO* mutants (**B** and **B'**), Gata3 protein is severely reduced in SGNs (**B**, circled) but is maintained in the cochlear duct (**B**, arrow). Mafb expression is also diminished (**B'**). In contrast, Mafb expression is restored in *Gata3^CKO;Mafb^CE* SGNs (**C'**), despite the loss of Gata3 (**C**). (**D-F'**) Confocal stacks from P15 control (**D** and **D'**), *Gata3^CKO* (**E** and **E'**) and *Gata3^CKO;Mafb^CE* (**F** and **F'**) cochlear whole-mounts double stained for GluR2 (green) and CtBP2 (red). The number of GluR2 and CtBP2 puncta is decreased in *Gata3^CKO* mice (**E** and **E'**) compared to controls (**D** and **D'**). Pre- and post-synaptic puncta are partially recovered in *Gata3^CKO;Mafb^CE* mice (**F** and **F'**). (**G**) Quantification of GluR2 puncta and ribbon number per IHC in control, *Gata3^CKO* and *Gata3^CKO;Mafb^CE* cochleae at P15. The number of GluR2 and CtBP2 puncta is decreased in *Gata3^CKO* cochleae but partially restored in *Gata3^CKO;Mafb^CE* cochleae when compared to controls. ***: $p<0.0005$. Scale bar in **C'** is 50 μm (**A–C'**); in **F'** is 5 μm (**D–F'**).
The following figure supplements are available for figure 8:

**Figure supplement 1**. Restoring Mafb expression in *Neurog1-Cre;Gata3^CKO* SGNs partially rescues the number of IHC ribbon synapses.

---

are master regulators, which induce multiple parallel changes in gene expression that reinforce cell fate and promote differentiation. The endpoint of the cascade is the activation of so-called terminal selectors, which induce batteries of genes essential for that neuron's functional properties. Terminal selectors work combinatorially, with multiple transcription factors cooperating to regulate sets of terminal effector genes that are necessary for each neuron's mature identity, such as neurotransmitters, channels, and synaptic adhesion molecules (*Hobert, 2011*). For example, the ETS-factor Ast-1 activates expression of genes essential for dopamine synthesis and transport in dopaminergic neurons (*Hobert, 2011*), acting together with Dlx and Pbx-family transcription factors (*Doitsidou et al., 2013*). Identifying analogous terminal selectors in vertebrates has been more challenging, due to the increased complexity of neuronal phenotypes, as well as the difficulty in defining discrete regulatory elements. Hence, the only established examples are for easily recognized features of mature neurons, such as the control of serotonin production by the transcription factor Pet1 (*Liu et al., 2010*).

Our results place Mafb at the bottom of a transcriptional network that controls the differentiation of neurons dedicated to the sense of hearing. At the top of this hierarchy are *Neurogenin1* and *NeuroD1* that control the generation of all inner ear neurons. Subsequently, *Gata3* expression is selectively enriched in SGNs (*Lawoko-Kerali et al., 2004*; *Lu et al., 2011*) and acts both to promote the auditory fate and to coordinate multiple aspects of SGN differentiation (*Appler et al., 2013*; *Duncan and Fritzsch, 2013*; *Luo et al., 2013*). Our findings add *Mafb* as a critical player downstream to Gata3: Gata3 is required for *Mafb* expression and *Mafb* is sufficient to rescue synapses in *Gata3^CKO* mutants. Hence, Mafb provides a new example of a transcription factor that can direct post-synaptic differentiation, apparently in a cell-type specific manner.

Mafb's restricted effects contrast with those of Gata3, which is necessary for multiple aspects of SGN development (*Appler et al., 2013*; *Duncan and Fritzsch, 2013*; *Luo et al., 2013*). This raises the important question of what other transcription factors might act downstream of Gata3. One attractive possibility is that combinatorial activity with other large Maf factors contributes to SGN differentiation. MafA and c-Maf are also highly expressed in SGNs (*Lu et al., 2011* and W-MY and LVG, unpublished observation), and both have been shown to play important roles in the terminal differentiation of neurons in the dorsal root ganglia (*Bourane et al., 2009*; *Wende et al., 2012*). Thus, there may be a Maf code for SGN terminal differentiation, with each factor directing a different feature required for mature function.

Mafb's potent effects on the late stages of SGN differentiation raise the intriguing possibility that Mafb may serve as a terminal selector in the vertebrate nervous system. Terminal selectors are typically recognized by two consistent characteristics: the ability to activate cell-type appropriate gene expression during terminal differentiation and an ongoing role in the maintenance of critical genes in the mature neuron (*Hobert, 2011*). Consistent with this definition, Mafb is not expressed until late in differentiation, just a day before functional synapses can first be detected (*Marrs and Spirou, 2012*), suggesting that Mafb may activate expression of terminal effectors that drive synapse formation. Additionally, Mafb is both necessary and sufficient for the acquisition of a key feature of mature SGN function, that is the ability to receive information from IHCs. Whether Mafb plays an ongoing role in SGNs remains to be determined, but its persistent expression in adults fulfills another important criterion.

Notably, the output of any one terminal selector is strongly influenced by cellular context, so individual terminal selectors can also be expressed in other subtypes of neurons and not all neurons of the same subtype depend on the same set of terminal selectors (*Doitsidou et al., 2013*). Similarly, the absence of Mafb from vestibular ganglion neurons, which also form ribbon synapses, suggests that different sets of transcription factors may act in SGNs and VGNs to control post-synaptic differentiation. Additional support for a terminal selector function for Mafb awaits identification of direct target genes, studies that may ultimately reveal effects on other auditory-specific features of SGN function.

Our findings raise the possibility that Mafb's effects might extend to other neuronal populations. Indeed, Mafb is expressed by a handful of other neuronal subtypes and could play a similar role in the emergence of their cell-type specific properties, including but not limited to synaptic differentiation. For instance, in addition to being expressed in SGNs, Mafb is also expressed in cochlear nucleus bushy cells (*Saul et al., 2008*), which receive synaptic input from SGNs. Similarly, Mafb defines subsets of cortical interneurons (*Wang et al., 2010*) and a subpopulation of rhythmogenic neurons within the pre-Bötzinger complex (*Blanchi et al., 2003*). Currently, the specific function of Mafb in other neuronal populations remains obscure, due both to the lethality of null mutants and the lack of tools for detecting subtle synaptic defects in other regions of the nervous system. With the availability of the new conditional alleles reported here, it will be important to investigate Mafb's activities in other regions of the nervous system in the future. In addition, our results suggest that similar late acting transcription factors may play analogous roles in other subtypes of neurons, functioning after synaptic targeting to ensure that each neuron elaborates the correct type of synapse necessary for proper circuit function.

Mafb's ability to direct auditory synaptogenesis even in pathological situations opens up the exciting possibility of harnessing this power to repair the damaged cochlea. One of the earliest effects of acoustic overexposure is the loss of synapses followed by withdrawal of SGN terminals. The SGNs subsequently undergo a long slow degeneration, ultimately resulting in significant hearing loss long after the original injury (*Kujawa and Liberman, 2009*). Our results suggest that stimulation of Mafb might be sufficient to encourage damaged neurons to re-establish connections with IHCs. Moreover, the specificity of Mafb's effects implies that synapses could be restored without inadvertently interfering with other aspects of SGN function. Importantly, we find that Mafb is still present in the adult cochlea (data not shown), largely sequestered in the cytoplasm. This observation emphasizes the need for further characterization of the post-translational modifications that affect the localization and transactivation activity of Mafb. Ultimately, drugs that enhance translocation of Mafb into the nucleus might offer a new treatment for deafness. Similarly, understanding how the Gata3–Mafb cascade is deployed will be critical for the development of stem cell-based replacement therapies. Indeed, recent studies have shown that functional dopaminergic and spinal motor neurons can be directly generated from somatic cells by ectopic expression of lineage specific transcription factors (*Caiazzo et al., 2011*; *Son et al., 2011*), thereby avoiding the need for surgical implantation of stem cells. With the discovery of transcription factors such as Neurogenin1, NeuroD1, Gata3, and Mafb, such an approach may eventually be possible in the cochlea.

## Materials and methods

### Generation of *Mafb^flox* and *Gt(ROSA)26Sor^CAG-lsl-Mafb* (*Mafb^CE*) mice

A linker sequence containing a *loxP* sequence and a SwaI restriction site for Southern blot genotyping was inserted into the middle of a 5.98 kB Mafb genomic fragment (corresponding to the sequence of 2260–8240 bp in *Figure 2A*) to create the Mafb 5′ targeting arm. Another linker sequence containing a BamHI restriction site for Southern blot genotyping was ligated to the 5′ end of a 1.07 kB Mafb genomic fragment (corresponding to the sequence of 8241–9312 bp in *Figure 2A*) to create the Mafb 3′ targeting arm. These arms were cloned into the MfeI/NheI sites and XhoI/NotI sites of a modified pBlueScript II SK+ plasmid containing a *loxP* sequence and an *frt*-flanked *Pgk-Neo* cassette (4600C conditional targeting vector) to generate the floxed Mafb targeting construct. This construct was electroporated into J1 ES cells (derived from 129S4/SvJae strain) and selected under G418 for 1 week. 395 ES cell clones were screened for correct recombination by Southern blot using external 5′ and 3′ probes (5D and 3C, 1212–2105 and 9930–11,019 bp in *Figure 2A*), and two successfully recombined clones were injected into blastocysts to make *Mafb^flox-neo* mice. These mice were crossed to a global *FLPe* driver [Tg(ACTFLPe), a gift from Dr Susan Dymecki, Harvard Medical School] (*Rodriguez et al., 2000*) to remove the *Pgk-Neo* cassette and generate *Mafb^flox* mice.

A 50 bp sequence upstream of Mafb translational initiation site and the Mafb coding sequence were cloned into the AscI site of a modified CTV plasmid (Addgene, Cambridge, MA) in which tdTomato replaces eGFP (Ctd plasmid). The Ctd–Mafb targeting construct was electroporated into J1 ES cells and clones were screened for correct recombination by PCR using two primer pairs amplified the 5′ and 3′ homology regions of the *ROSA26* locus (forward-CGCCTA AAGAAGAGGCTGTG and reverse-GGGCG-TACTTGGCATATGAT for a 1472 bp 5′ homology region; forward-GCCTCGACTGTG CCTTCTAG and reverse-CCATTCTCAGTGGCTCAACA for a 4886 bp 3′ homology region). Two successfully recombined clones were injected into blastocysts to make *Gt(ROSA)26Sor*$^{CAG-lsl-Mafb}$ (*Mafb*$^{CE}$) mice.

## Mouse strains, genotyping and staging

*Mafb*$^{flox}$ mice were PCR genotyped using *Mafb-loxP* forward (CCTCAACGGCTTCGGGGCTCCTC) and reverse (CGCTCTCCGACTCTTTGGCTCTA) primers with 5% dimethyl sulfoxide in the reaction (purple arrows in **Figure 2A** indicate the primer locations; wt, 264 bp; *Mafb*$^{flox}$, 310 bp). *Mafb*$^{CE}$ mice were PCR genotyped using *tdTomato* forward (CTGTTCCTGTACGGCATGG) and reverse (GGCATTAAAGCAGCGTATCC) primers and control forward (AAGGGAGCTGCAGTGGAGTA) and reverse (CCGAAAATCTGTGGGAAGTC) primers (wt, 297 bp; *Mafb*$^{CE}$, 196 bp). The following mouse strains were used and PCR genotyped as described previously: a null GFP knock-in allele of *Mafb* (*Mafb*$^{GFP}$) (**Moriguchi et al., 2006**), two floxed Gata3 alleles: *Gata3*$^{fl}$ (**Zhu et al., 2004**) and *Gata3*$^{Ho}$ (**Pai et al., 2004**), a null tauLacZ knock-in allele of *Gata3* (*Gata3*$^{tauLacZ}$) (**van Doorninck et al., 1999**), β-actin-Cre [*Tg(ACTB-cre)*] (the Jackson Laboratory, Bar Harbor, ME), *Foxg1-Cre* (**Hebert and McConnell, 2000**), *Neurog1-Cre* (**Quinones et al., 2010**), *Bhlhe22-Cre* (**Ross et al., 2010**), *Thy1-YFP* (YFP-12) (**Fu et al., 2010**), and the tdTomato reporter [Ai14; *Gt(ROSA)26Sor*$^{tm14(CAG-tdTomato)Hze}$] (Allen Institute for Brain Science, Seattle, WA). For *Mafb*$^{CKO}$ mice, all mutants contained a null and a floxed allele (*Neurog1-Cre;Mafb*$^{flox/GFP}$) and all controls were littermates heterozygous for *Mafb* (*Neurog1-Cre;Mafb*$^{flox/+}$, *Neurog1-Cre;Mafb*$^{GFP/+}$ or *Mafb*$^{GFP/flox}$) except for two *Mafb*$^{flox/flox}$ animals used as controls for the ABRs due to the lack of heterozygous littermates. For *Gata3*$^{CKO}$ mice, all mutants contained a null and a floxed allele (*Bhlhe22-Cre;Gata3*$^{flox/tauLacZ}$), and all controls were heterozygous littermates (*Gata3*$^{flox/tauLacZ}$). For staging, noon on the day that the vaginal plug was observed was counted as embryonic day 0.5 (E0.5). For postnatal collection, the day of birth was considered P0. Embryos and pups of both sexes were used. Animals were maintained and handled in compliance with a protocol approved by the Institutional Animal Care and Use Committee at Harvard Medical School.

## In situ hybridization

E13.5 embryo heads were fixed in 4% paraformaldehyde (PFA) in PBS overnight at 4°C. P6 mice were perfused with 4% PFA in PBS, and the inner ears were dissected out and post-fixed in 4% PFA in PBS for 2 hr at 4°C. All tissue was sucrose protected and then frozen in a Neg50 frozen section medium (Richard-Allan Scientific, Kalamazoo, MI). 12-μm thick cryosections from pairs of control and *Mafb*$^{CKO}$ animals were collected on the same Superfrost Plus slide (VWR International, Radnor, PA). Non-radioactive in situ hybridization was performed as described (**Abraira et al., 2008**). A detailed protocol is available at http://goodrich.med.harvard.edu/resources/resources_protocol.htm. The template DNA used for ribo-probe synthesis was PCR amplified from cDNA clones (*Mafb*: BC038256; *GluR2*: BC048248; *PSD95*: BC014807) using the following primers: *Mafb*, CCGGAATTCCAGTCCGACTGAACCGAAGACC and CCCAAGCTTCTCAGGAGAGGAGGGGCTGTCG (GenitoUrinary Development Molecular Anatomy Project); *GluR2*, TGTATCCTTCATCACACCAAGC and GTCATCACTTGGACAGCATCAT (Allen Brain Atlas); *PSD95*, CGATTACCACTTTGTCTCCTCC and AAGAAAGGCTAGGGTACGAAGG (Allen Brain Atlas). For the semi-quantitative analysis of in situ hybridization, images were converted to 8-bit gray-scale. A mask corresponding to a region of 50 × 50 μm was superimposed over SGNs in the middle turn of the cochlea. The mean signal intensity of the region with the background removed was calculated using ImageJ (National Institutes of Health, Bethesda, MD).

## Western blot analysis

E16.5 whole embryos or postnatal cochleae were homogenized in 25 mM Tris, pH 7.5 containing 95 mM NaCl, 5 mM EDTA, 2% SDS and 1 mM Pefabloc (Roche, Penzberg, Germany). Western blot analysis was performed using standard protocols. Rabbit anti-Mafb (1:1000, NB600-266; Novus Biologicals, Littleton, CO) and mouse anti-β-actin (1:5000, ab8226; Abcam, Cambridge, England) primary antibodies were used.

## Paintfilling

Paintfilling of E14.5 mouse embryos was performed by injecting White All Purpose Correction Fluid (Sanford Corporation, Oak Brook, IL) into the cochlea as described previously (*Abraira et al., 2008*).

## Plastic section preparation and SGN counts

Araldite sections were prepared and SGNs were counted as described previously (*Kujawa and Liberman, 2009*) with two minor modifications. Sections were cut at 20 µm instead of 40-µm thickness, and a mask corresponding to a region of 60 × 60 µm was superimposed on the P3 image instead of the 90 × 60 µm mask used with the P15 ganglion due to the smaller size of the P3 ganglion.

## Immunofluorescence

Embryonic heads were fixed directly in 4% PFA in PBS overnight at 4°C. Postnatal mice were anesthetized with ketamine and xylazine and then intravascularly perfused with 4% PFA in PBS. Inner ears were dissected and then the round and oval windows were opened to permit flushing of 4% PFA through the scalae, followed by post-fixation in 4% PFA overnight at 4°C. For mice older than P7, inner ears were decalcified in 120 mM EDTA in PBS at room temperature for 2 days. For tissue sections, tissues were stepped through 10, 20, and 30% sucrose in PBS, embedded in NEG 50 (Richard-Allan Scientific), and cryosectioned at 14 µm. The sections were blocked 1 hr at room temperature in a solution containing 5% normal donkey serum and 0.3% Triton X-100 in PBS. The sections were then incubated overnight at 4°C in primary antibodies diluted in the blocking solution. Alexa Fluor-conjugated secondary antibodies (Life Technologies, Carlsbad, CA) were used for signal detection. Rabbit anti-Mafb (1:500, NB600-266; Novus Biologicals), mouse anti-TuJ1 (1:2000, MMS-435P; Covance, Dedham, MA), and goat anti-Gata3 (1:100, AF2605; R&D Systems, Minneapolis, MN) primary antibodies were used.

For cochlear whole-mount staining, fixed cochleae were dissected into three pieces corresponding to 2.8–11.3 kHz (apex), 11.3–32 kHz (middle), and 32 kHz–64 kHz (base) regions in the cochlea (*Muller et al., 2005*). The microdissected pieces were blocked in PBS with 1%Triton X-100 and 5% normal donkey serum for 1 hr at room temperature and then incubated in primary antibodies diluted in blocking solution at 37°C for 20 hr. Alexa Fluor-conjugated secondary antibodies were used for signal detection. Chicken anti-NF-H (1:3000, AB5539; Millipore, Billerica, MA), rabbit anti-TuJ1 (1:1000, MRB-435P; Covance), goat anti-Choline Acetyltransferase (1:200, AB144P; Millipore), mouse anti-synaptophysin 1 (1:200, 101 011, Synaptic Systems, Goettingen, Germany), rabbit anti-Vesicular Acetylcholine Transporter (1:1000, ab68984, Abcam), and mouse anti-PSD95 (1:500, MABN68; Millipore) primary antibodies were used. For CtBP2 and GluR2 double staining, cochlear pieces were permeabilized in 30% sucrose for 20 min and stained as described previously (*Furman et al., 2013*). Rabbit anti-Myo7A (1:200, 25-6790; Proteus BioSciences, Ramona, CA), mouse IgG1 anti-CtBP2 (1:200, 612044; BD Transduction Laboratories, San Jose, CA) and mouse IgG2a anti-GluR2 (1:1000, MAB397; Millipore) primary antibodies were used.

## Confocal imaging, 3D reconstruction and synaptic counts

Cochlear lengths were measured for each microdissected piece and a cochlear frequency map was determined as described previously (*Muller et al., 2005*). Confocal z-stacks from the selected cochlear region were obtained on an Olympus FluoView FV1000 (Tokyo, Japan) using a 40X (NA:1.30) or a 60X (NA:1.40) oil-immersion objectives. A 512 × 512 binary image was acquired at optimal step size in the Z axis (0.55 µm for 40X and 0.45 µm for 60X). For confocal imaging of CtBP2 and GluR2 staining, z-stacks from the 16 kHz region were obtained on an Olympus FluoView FV1000 using a 60X (NA:1.40) oil-immersion objective and 2.5X digital zoom. A 512 × 512 (pixel size = 0.165 µm in x and y) binary image was acquired at 0.45-µm step size in the Z axis, resulting in an image containing approximately 10 IHCs. Care was taken to minimize pixel saturation in each image stack and to acquire control and mutant images using the same laser power and high voltage value of photomultiplier.

For quantification of GluR2 immunofluorescence of P6 cochleae, a maximum projection of the stacks containing GluR2 immunofluorescence in a region of 10 IHCs was obtained using Olympus Fluoview software. A mask corresponding to a region of 80 × 20 µm was superimposed on the image to cover the entire synaptic poles of 10 IHCs. Fluorescence intensity was calculated minus the background using ImageJ (National Institutes of Health). *Mafb*^CKO^ or *Mafb*^CE^ fluorescence intensity is expressed as a percentage of value from intensity of littermate controls.

For 3D reconstruction and counting of GluR2 and CtBP2 puncta, image stacks were exported to Amira imaging-processing software (Visualization Sciences Group, Burlington, MA), where three-dimensional reconstruction of IHCs, synaptic ribbons and GluR2 puncta were produced. The number of ribbons and GluR2-positive puncta was divided by the total number of IHCs, as revealed by Myo7A and the presence of CtBP2-positive nuclei (including fractional estimates, when necessary, at each end of the image stack). For measurement of the three-dimensional distance to nearest neighbor, x, y, and z coordinates of all ribbons were exported from Amira to Excel (Microsoft, Redmond, WA). Each ribbon was assigned manually to an IHC according to the x, y, and z coordinates of a 3D rendering of the Myo7A 'isosurfaces'. The inter-ribbon distance within an IHC was then calculated using each ribbon's x, y, and z coordinates.

## Electron microscopy

Animals were anesthetized and then intravascularly perfused with 0.1M sodium cacodylate buffer pH 7.4 containing 2.5% glutaraldehyde and 1.5% paraformaldehyde. Inner ears were removed and the round and oval windows were opened to permit flushing of fix through the scalae followed by postfixation overnight at 4°C. The cochlear coils were dissected, separated into three pieces and postfixed in 2% osmium tetroxide at room temperature for 1 hr. The cochlear pieces were dehydrated and embedded in Epon resin. 80-nm ultrathin sections from the center region of the middle turn were counterstained with uranyl acetate and lead citrate, and observed with a JEOL 1200EX electron microscope (Akishima, Japan).

## Measurements of auditory function

For measuring of auditory brainstem recordings (ABRs) and distortion product otoacoustic emissions (DPOAEs), recordings were performed on the right ears of 5- to 7-week-old anesthetized mice in a soundproof chamber maintained at 32°C as described previously (*Buran et al., 2010*). Each *Mafb*[CKO] mutants was paired with a control littermate during each round of recordings (16 mice in each group). Average ABR waveforms were plotted using a script written in MATLAB (MathWorks, Natick, MA) by Dr Ann E Hickox in the laboratory of Dr Charles Liberman (Massachusetts Eye and Ear Infirmary, Boston, MA).

## Statistics

Data were analyzed with Excel (Microsoft) and GraphPad Prism (GraphPad, La Jolla, CA). Student's *t*-test was used to determine if two sets of data are significantly different from each other. Difference in means were considered significant if $p < 0.05$. The results are expressed as means ± SEMs unless otherwise noted.

## SGN organotypic explants

SGNs were acutely dissected from P3 and P4 mice. Following rapid decapitation, cochleae were excised from the temporal bone, and bathed in sterile MEM with glutamax (Gibco # 41,090, Grand Island, NY) supplemented with 10 mM HEPES (Sigma, St. Louis, MO) and 25 mg ampicillin (Sigma), pH 7.4. SGN explants were divided into halves along the apical-basal axis and mounted flat on glass cover slips. SGN organotypic explants were incubated at 37°C in a humidified incubator (5% $CO_2$) for 1–2 hr, followed by acute electrophysiological studies. All SGN explants were used within 4–5 hr of dissection.

## Electrophysiology

The whole-cell, tight-seal technique was used in voltage- and current-clamp modes to record from identified SGN cell bodies. SGN organotypic explants were placed into a custom-made recording chamber and viewed under Zeiss Axioskop FS upright microscope (Oberkochen, Germany) equipped with a 63× water-immersion lens and differential interface contrast optics. SGN explants were bathed in standard external solution that contained (in mM): 137 NaCl, 0.7 $NaH_2PO_4$, 5.8 KCl, 1.3 $CaCl_2$, 0.9 $MgCl_2$, 5.6 D-glucose, 10 HEPES, amino acids (1:50, 11130; Gibco), vitamins (1:100, 11120; Gibco); pH 7.4 (NaOH), 303 mOsmol/kg. Recording pipettes (3–5 MΩ) were pulled from R-6 soda lime capillaries (King Precision Glass, Claremont, CA), using a two-stage vertical pipette puller (PC-10; Narishige, Tokyo, Japan). Recording pipettes were filled with standard internal solution that contained (in mM): 135 KCl, 2.5 $MgCl_2$, 2.5 $K_2$-ATP, 5.0 HEPES, 5.0 EGTA, 0.1 $CaCl_2$; pH 7.4 (KOH), 283 mOsmol/kg.

Electrophysiological data from SGNs were recorded using an Axopatch 200B amplifier (Molecular Devices, Palo Alto, CA). Signals were filtered at 1 kHz with a low pass Bessel filter, and digitized at ≥20 kHz using 12-bit acquisition system Digidata 1332 (Axon Instruments, Union City, CA) and pClamp 9.0 software (Molecular Devices). Currents were recorded immediately after the cell membrane was broken through at giga-ohm (GΩ) seal and the series resistance ($R_s$) and membrane capacitance ($C_m$) were corrected. Compensated residual $R_s$ was below 7 MΩ on average. All electrophysiological recordings were performed at room temperature (22-24°C). Offline data analysis was performed using OriginPro 7.5 (Origin Lab, Northampton, MA) and reported as mean ± SD. Liquid junction potentials (−4 mV) were adjusted offline for all membrane potentials.

## Acknowledgements

We thank the Transgenic Core Facility (Brigham and Women's Hospital) and the Mouse Gene Manipulation Facility of the Boston's Children's Hospital Intellectual and Developmental Disabilities Research Center (IDDRC), which is supported by NIH P30-HD18655, for help in transgenic animal production. We are grateful to D Corey and members of his laboratory for sharing equipment; MC Liberman, A Furman, A Hickox, and C Miller (Eaton Peabody Laboratory, Massachusetts Eye and Ear Infirmary) for assistance with analysis of ribbon synapses, hearing function and SGN counting; M Gordon for genotyping assistance; A Tucker for in situ hybridization; L Trakimas for assistance with ultrathin section preparation; the Harvard Neurobiology Imaging Facility; and C-Y Lu for assistance with figure preparation.

## Additional information

### Funding

| Funder | Grant reference number | Author |
|---|---|---|
| National Institutes of Health | R01 DC009223 | Lisa V Goodrich |
| March of Dimes | | Lisa V Goodrich |
| Hellman Family Foundation | | Lisa V Goodrich |
| The Bertarelli Foundation | | Lisa V Goodrich |
| National Institutes of Health | F32 DC012695 | Wei-Ming Yu |
| Alice and Joseph Brooks Foundation | | Wei-Ming Yu |
| Harvard Mahoney Neuroscience Institute | | Wei-Ming Yu |
| Stuart H.Q. and Victoria Quan Foundation | | Jessica M Appler |
| Lefler Family Foundation | | Allison M Nishitani |
| National Institutes of Health | R01 DC005439 | Jeffrey R Holt |
| Action on Hearing Loss | R58 | Lisa V Goodrich |

The funders had no role in study design, data collection and interpretation, or the decision to submit the work for publication.

### Author contributions

W-MY, Conception and design, Acquisition of data, Analysis and interpretation of data, Drafting or revising the article, Contributed unpublished essential data or reagents; JMA, Conception and design, Acquisition of data, Contributed unpublished essential data or reagents; Y-HK, Acquisition of data, Analysis and interpretation of data, Drafting or revising the article; AMN, Acquisition of data, Drafting or revising the article; JRH, LVG, Conception and design, Analysis and interpretation of data, Drafting or revising the article

### Ethics

Animal experimentation: This study was performed in strict accordance with the recommendations in the Guide for the Care and Use of Laboratory Animals of the National Institutes of Health. All of the animals were handled according to the approved institutional animal care and use committee (IACUC) protocol (#03611) of Harvard Medical School.

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
