## [Decision Letter]

Thank you for sending your work entitled “A Gata3-Mafb transcriptional network directs post-synaptic differentiation in synapses specialized for hearing” for consideration at *eLife*. Your article has been favorably evaluated by a Senior editor and 3 reviewers, one of whom is a member of our Board of Reviewing Editors.

The Reviewing editor and the other reviewers discussed their comments before we reached this decision, and the Reviewing editor has assembled the following comments to help you prepare a revised submission.

Yu et al. generated conditional knock-out and conditional knock-in mice and demonstrated that transcription factor Mafb, downstream of GATA3, is required for proper postsynaptic differentiation of spiral ganglion neurons (SGNs). In a *Mafb* conditional mutant, SGNs have smaller PSD and reduced GluR2 clustering (puncta), and these defects in turn lead to reduced presynaptic ribbons in hair cells. Expressing Mafb earlier in SGN development caused earlier clustering of GluR2, but did not affect the mature patterns of hair cell-SGN synapses. Electrophysiology recording confirmed that defects in synapse formation caused abnormal auditory responses.

The experiments were all performed carefully and the data are convincing. Given that very little is known about the transcriptional control of synapse formation in general, and in auditory development in particular, we are enthusiastic in receiving a revised manuscript from the authors that will address the following three critiques. The first two are related to interpretations of the data and the third requires additional experiments.

1) We are not convinced by the authors’ interpretation that Mafb drives cell-type specific features of terminal differentiation, i.e., it is a terminal selector.

Specifically, the authors described the specific features as the ribbon synapses: the large PSD and the presynaptic ribbons. However, these ribbon synapses are also present in the vestibular hair cell-neuron synapses, and Mafb is not expressed in the vestibular ganglion neurons. Thus, one does not need Mafb to make ribbon synapses. Moreover, overexpression of Mafb did not give rise to more ribbon synapses; it only had a transient effect on GluR2 clustering. Thus, Mafb could not “drive” ectopic specialized synapse formation, and therefore it is unlikely to be a terminal selector.

One could imagine a scenario that transcriptional targets of Mafb are synaptogenetic adhesion molecules required to bring pre- and post- synaptic membrane together; while the hair cells have intrinsic capacity to form ribbons, once the pre- and the post- synaptic membranes are opposed to each other, the hair cells can drive the postsynaptic differentiation in SGN neurons. In this scenario, the “driver” of terminal differentiation and ribbon synapse formation is the hair cells, not genes in SGN neurons.

In fact, the phenotype shown in Figure 4, the mis-alignment (or mis-match) of pre- and post synaptic puncta is consistent with the idea that some synaptogenic adhesion process was abnormal in *Mafb* mutant, and as a consequence synapses fail to further develop/differentiate.

2) We are not entirely convinced by the conclusion “Precocious expression of Mafb accelerates afferent synapse development.” The genetic strategy enables the authors to express Mafb at E12.5, as opposed to the normal onset between E13.5-E15.5. However, the “precocious” postsynaptic development was examined at P6, or nearly two weeks after the normal onset of expression. This effect could well be caused by overexpression of Mafb, rather than precocious expression of Mafb.

3) The reviewers all agree that identifying the transcriptional targets of Mafb to get at the mechanisms underlying the defects of hair cell-SGN synapse formation is an important next step for subsequent studies. However, since Mafb is a transcription factor, and the authors did show reduced GluR2 protein puncta and reduced PSD size in the Mafb mutant, the authors should at least do a few in situ hybridization analyses to examine whether the mRNA of GluR2 and PSD components are reduced.

---

## [Author Response]

*1) We are not convinced by the authors’ interpretation that Mafb drives cell-type specific features of terminal differentiation, i.e., it is a terminal selector*.

*Specifically, the authors described the specific features as the ribbon synapses: the large PSD and the presynaptic ribbons. However, these ribbon synapses are also present in the vestibular hair cell-neuron synapses, and Mafb is not expressed in the vestibular ganglion neurons. Thus, one does not need Mafb to make ribbon synapses. Moreover, overexpression of Mafb did not give rise to more ribbon synapses; it only had a transient effect on GluR2 clustering. Thus, Mafb could not “drive” ectopic specialized synapse formation, and therefore it is unlikely to be a terminal selector*.

*One could imagine a scenario that transcriptional targets of Mafb are synaptogenetic adhesion molecules required to bring pre- and post- synaptic membrane together; while the hair cells have intrinsic capacity to form ribbons, once the pre- and the post- synaptic membranes are opposed to each other, the hair cells can drive the postsynaptic differentiation in SGN neurons. In this scenario, the “driver” of terminal differentiation and ribbon synapse formation is the hair cells, not genes in SGN neurons*.

*In fact, the phenotype shown in*
Figure 4*, the mis-alignment (or mis-match) of pre- and post synaptic puncta is consistent with the idea that some synaptogenic adhesion process was abnormal in Mafb mutant, and as a consequence synapses fail to further develop/differentiate*.

We agree that our evidence does not demonstrate conclusively that Mafb is a terminal selector in SGNs. However, we feel that this is an attractive idea to consider, since Mafb is activated during terminal differentiation, is required for a key step in SGN differentiation, and continues to be expressed in adult SGNs. The fact that Mafb is absent from a different population of neurons that also makes ribbon synapses does not rule out a terminal selector function. In fact, terminal selectors work combinatorially and cooperatively. Hence, an individual terminal selector may be present in a variety of cell types but can only activate a particular set of effector genes when other transcription factors are also present. Moreover, the same set of effector genes can be regulated by different combinations of transcription factors, as shown recently for regulation of dopamine pathway genes in different subtypes of dopaminergic neurons in *C. elegans* (11). Similarly, vestibular ganglion neurons may employ different combinations of transcription factors for regulation of post-synaptic differentiation, possibly including other Maf factors. Along the same lines, Mafb may require the presence of additional factors in order to alter synapse number permanently when misexpressed. Alternatively, pruning mechanisms may overcome the synapse-promoting effects of Mafb.

Clearly, more work is needed to determine whether Mafb is indeed a terminal selector, beginning with identification of target genes in SGNs. With regards to the scenario presented, it is certainly possible that Mafb induces expression of a synaptogenic adhesion molecule, but this alone would not be sufficient to reject the terminal selector possibility, particularly if ongoing expression of that molecule is necessary for maintenance of the mature synapse. We hesitate to make any conclusions for or against the proposed model based on the data shown in Figure 4, as there are also other interpretations. We have rewritten the Discussion to make these points clearer without overstating a possible role for Mafb as a terminal selector.

*2) We are not entirely convinced by the conclusion “Precocious expression of Mafb accelerates afferent synapse development.” The genetic strategy enables the authors to express Mafb at E12.5, as opposed to the normal onset between E13.5-E15.5. However, the “precocious” postsynaptic development was examined at P6, or nearly two weeks after the normal onset of expression. This effect could well be caused by overexpression of Mafb, rather than precocious expression of Mafb*.

This point is well taken. We had focused on the change in timing because Western blot analysis did not reveal an obvious increase in the amount of Mafb at postnatal stages, when Mafb is already present in abundance (data not shown). Unfortunately, we were not able to measure Mafb levels earlier, as it is technically challenging to collect sufficient spiral ganglia from mice with different genotypes at embryonic stages for Western Blots. However, as one would predict given the earlier onset of expression, immunostaining between E12.5 to E16.5 indicated that the level of Mafb proteins is higher in *Mafb^CE^* SGNs when compared to control SGNs at the same stage. The cumulative effects of increased levels of Mafb over this long period of time could indeed explain the accelerated changes in synapse development we see at P6. We have made changes in the text accordingly.

*3) The reviewers all agree that identifying the transcriptional targets of Mafb to get at the mechanisms underlying the defects of hair cell-SGN synapse formation is an important next step for subsequent studies. However, since Mafb is a transcription factor, and the authors did show reduced GluR2 protein puncta and reduced PSD size in the Mafb mutant, the authors should at least do a few* in situ *hybridization analyses to examine whether the mRNA of GluR2 and PSD components are reduced*.

While identification of the complete set of Mafb target genes remains an important goal for the future, we agree it is important to address whether the mRNA levels of GluR2 and PSD components are reduced in *Mafb^CKO^* mutant neurons. We have performed the in situ hybridization of GluR2 and PSD95 in P6 control and *Mafb^CKO^* SGNs as suggested. We found that *Mafb^CKO^* SGNs show a similar level GluR2 and PSD95 mRNAs when compared to control SGNs. This indicates that the reduced number of GluR2 puncta and formation of smaller PSDs in mutant animals cannot be explained by decreased expression of GluR2 or other PSD components. These results were added to the text and in Figure 4—figure supplement 2.